# An Irreversibility-Based Criterion to Determine the Cost Formation of Residues in a Three-Pressure-Level Combined Cycle

**DOI:** 10.3390/e22030299

**Published:** 2020-03-05

**Authors:** Lugo-Méndez Helen Denise, Torres-González Edgar Vicente, Castro-Hernández Sergio, Salazar-Pereyra Martín, López-Arenas Teresa, Lugo-Leyte Raúl

**Affiliations:** 1Departamento de Procesos y Tecnología, Universidad Autónoma Metropolitana–Cuajimalpa, Av. Vasco de Quiroga 4871, Santa Fé 05348, Cuajimalpa, Ciudad de México, Mexico; hlugo@correo.cua.uam.mx (L.-M.H.D.); mtlopez@correo.cua.uam.mx (L.-A.T.); 2Departamento de Ingeniería de Procesos e Hidráulica, Universidad Autónoma Metropolitana–Iztapalapa, Av. San Rafael Atlixco 186, Vicentina 09340, Iztapalapa, Ciudad de México, Mexico; etorres@xanum.uam.mx (T.-G.E.V.); sch@xanum.uam.mx (C.-H.S.); 3Tecnológico de Estudios Superiores de Ecatepec, División de Ingeniería Mecatrónica e Industrial, Av. Tecnológico Esq. Av. Hank González, Valle de Anáhuac 55210, Ecatepec, Estado de México, Mexico; msalazar@tese.edu.mx

**Keywords:** exergetic cost analysis, cost formation of residues, irreversibilities, three-pressure-level combined cycle

## Abstract

In an energy system, it is important to identify the origin of residue formation in order to implement actions to reduce their formation or to eliminate them as well as to evaluate their impact on the production costs of the system. In the exergetic cost theory, although there are several criteria to allocate the cost formation of residues to the productive components, no unique indication on the best choice has been defined yet. In this paper, the production exergy costs are determined by allocating the residue cost formation to the irreversibilities of the productive components from which they originate. This criterion, based on the Gouy–Stodola theorem, is an extension of the criterion of entropy changes, and unlike this, it avoids the existence of a negative production cost. This criterion is applied to a combined cycle of three pressure levels, and the production exergy costs are compared with the criteria of entropy changes, distributed exergy, and entropy. The results of the proposed criterion are in agreement with the compared criteria.

## 1. Introduction

From an exergoeconomic point of view, the product formation of an energy system is carried out in the productive components and it is always accompanied by the formation of unintended remaining flows of matter or energy, called residues. They could be partially used in further processes or could become unusable or unwanted waste disposals [1]; for example, in a combined cycle power plant, the exhaust gases of the gas turbine are used favorably to generate steam and to increase the net output power. The residues are exergetic losses; they are partial or totally eliminated to the environment in dissipative components. The presence of dissipative components is essential for the proper functioning of a power plant, and its utility lies in interacting with other components, which in some cases allows the system to have a higher production or better efficiency, for example, the condenser of a steam cycle [2].

The exergoeconomic methods have traditionally placed emphasis on the cost formation process of products. However, every residue also has a cost formation process that must be identified to calculate the cost of all products correctly. It is necessary therefore to develop new techniques or to extend the existing ones that include the analysis of the formation process of residues and their allocation costs to the cost of products [1]. In an improved exergoeconomic analysis, the cost of residues, which includes their formation cost (cost of their exergy content) and their abatement cost (cost of the resources employed in their treatment), is not allocated to the dissipative components but to the productive components involved in their formation and, thus, to the final products, according to the productive structure of the system [2]. In this approach, the cost of a residue is decomposed into several costs, one for each component that has generated it, by means of so-called residue cost allocation ratios, and nevertheless, there is not a definitive way to determine these ratios [3]. Although there has been an advance in the development of criteria for the cost allocation of residues, this problem is still an open research line [4].

A common criterion to determine the residues cost allocation ratios for closed cycles, such as Rankine or refrigeration cycles, is based on the sum of the entropy changes of the working fluid in each productive process equalling the entropy saved on the dissipative process [5,6]. As mentioned by Torres et al. [2], this allocation criterion fails for open cycles like gas turbines. In the case of a simple gas turbine with a heat recovery steam generator, only a part of the entropy generated in the global process is saved and the environment is the dissipative component responsible for closing the cycle by reducing the remaining working fluid entropy to the dead state. In this criterion, a product resulting from a productive process that generates more entropy is more penalized than a product that originated from a productive process generating less entropy [7]. This criterion has been used in closed cycles, such as Rankine or refrigeration cycles, gas turbine cogeneration system [5,6,8,9,10,11,12], and an extraction-condensing steam turbine cogeneration system coupled with a multiple-effect thermal vapor compression desalination unit [13].

Torres et al. [2] highlight the importance of the residue cost formation process and present a methodology based on the symbolic exergoeconomics methodology to evaluate the costs and residue formation process of a one-pressure level combined cycle. The authors use the criterion proposed by Valero et al. [14] and define the residues cost allocation ratios as the residues cost distribution ratios of the residues. In this criterion, based on the productive structure and using the fuel–product–residue table as a starting point, the cost of the residue is divided into several costs, one for each component that has generated it. Torres et al. [2] apply this criterion to a simple combined cycle power plant, in which the exhaust gases and the waste heat dissipated in the condenser are the residues of the system. In this work, the cost formation of the exhaust gases is allocated only to the combustion chamber and the compressor while the cost of the other residue is allocated to components of the heat recovery steam generator and the pump of the steam cycle. Valero et al. [15] perform an exergoeconomic analysis of a cogeneration plant conformed by a steam power cycle with coal as a fuel, a cement production process, and a steam generation process with natural gas. In this cogeneration system, the ashes of the power cycle are sent to the cement production process. The residues of the system are the heat discarded in the condenser and the exhaust gases of the steam generator, and they are allocated to the productive components on the basis of the recirculation coefficients of residues.

Seyyedi et al. [16] propose a new approach based on the entropy distribution in the components and use the exergy and enthalpy fuel-product tables as input data. The residue cost allocation ratios combine then the exergy and enthalpy distribution coefficients of the productive components of which the products serve as resource to the dissipative components, and they are related to the entropy distribution through the dissipative components. The proposed criterion is applied to a combined cycle of one-level pressure with the heat discarded in the condenser and the exhaust gases released in a stack to the environment as the residues. The results are compared with the criteria of entropy changes and distributed exergy, and the authors conclude that the residue cost allocation ratios derived from their criterion are comprised between those obtained by the other two criteria. This criterion has been used to carry out an exergoeconomic analysis of a combined power cycle integrated with a high-temperature cooled-gas reactor, in which the gas turbine is a closed Brayton with helium gas as working fluid and the waste heat dissipated in the condenser is the only residue [17].

Agudelo et al. [3] carry out a study to allocate the residue cost to the productive components of a combined cycle of one pressure level and a cogeneration system with a gas turbine. The study is based on the mathematical formulation for the cost formation of residues, proposed by Torres et al. [2], and it defines the concept of “waste cost distribution ratio” as a means to determine the responsibility of any productive component to any generated residue. Recently, this criterion has been used by Gao et al. [18] and Zhang et al. [19] to conduct an exergy-based analysis of a coal-fired combined heat and power system with three residues, corresponding to exhaust gases, waste heat dissipated from the condenser, and ashes produced by the coal combustion.

In the works cited above, two approaches based on the application of the rules of the thermoeconomic theory are pursued to compute the production costs. In the first one, the set of costing equations is directly solved to compute the cost of the exergy flows, and in the second one, the production costs of productive components are calculated applying the symbolic exergoconomic methodology, using the fuel-product table as the starting point [2,8].

In this paper, the production costs are determined employing the first approach and allocating the residue cost formation to the irreversibilities of the productive components from which they originated. This criterion based on irreversibilities can be conceived as an extension of the criterion of entropy changes, and it is supported on the Gouy–Stodola theorem. The thermodynamic and exergy analysis of a three-pressure-level combined cycle plant is conducted in order to use this energy system as a case study to illustrate the application of the proposed criterion. The criterion based on irreversibilties is compared with respect to the criteria of entropy changes, exergy, and entropy recirculation; these three criteria are described and applied to the same combined cycle.

## 2. Thermodynamics of the Combined Cycle

The energy system Tuxpan II is a three-pressure-level combined cycle plant installed in the Tuxpan area (about 250 km northeast from Mexico City), State of Veracruz, Mexico, and has operated since 15 December 2001. The combined cycle plant is composed by two identical modules conformed by a Mitsubishi M501F3 gas turbine coupled to its respective heat recovery steam generator (HRSG). The steam generated in both HRSGs is supplied to a steam turbine. Each combined cycle module generates 427.25 MW. The physical model of the three-pressure-level combined cycle module is shown in Figure 1. In this section, the thermodynamic model of the plant is described through a set of equations, and from the solution of these equations, pressure, temperature, enthalpy, entropy, and exergy of flows are obtained.

### 2.1. Assumptions Made

The assumptions made in the thermodynamic study carried out in this paper are listed below:The combined cycle operates under steady-flow conditions.The air and combustion gas flows are assumed to behave ideally.The components of the combined cycle are taken into consideration as adiabatic.Kinetic and potential energy changes within each component are assumed to be negligible.The fuel physical exergy is not considered.Chemical exergy is neglected in the exergoeconomic analyses of all the combined cycle components.The operating conditions of the combined cycle, the dead state, the pinch-point difference temperatures, and the combustion parameters of the study case are presented in Table 1.

### 2.2. Gas Turbine

A gas turbine has an upstream rotating compressor coupled to a downstream turbine and a combustion chamber in between. A simple gas turbine operates on the thermodynamic cycle presented in the exergy–enthalpy diagram of Figure 2a, in which air entering the compressor at state g1 is compressed to some higher pressure at state g2. Leaving the compressor, air enters the combustion chamber, where combustion occurs by fuel injection. In the combustion process, a pressure drop occurs by the mixing, burning, and cooling phenomena. The exhaust gases leave the combustion system and enter the turbine; the hot gases are expanded to state g4 at a higher pressure than the air pressure at state g1 to generate the useful output power. Since the highest temperature and pressure reached in the gas turbine correspond to the turbine inlet state, g3, this state presents the maximum exergy.

In this analysis, it is assumed the gas turbine cycles of each module of the combined cycle operate identically. On the basis of the assumption that the air and combustion gases behave as ideal gases, mass and energy conservation laws are applied to each component of the gas turbine to deduce the following gas turbine thermodynamic performance equations:Heat supplied to the gas turbine
(1)$$q_{H_{GT}}=c_{Pair}T_{g1}\left[\left(1+far\right)\frac{c_{Pcg}}{c_{Pair}}y-1-\frac{\pi_{C}^{x_{air}}-1}{\eta_{C}}\right]$$
where $x_{air}=\frac{R_{air}}{c_{Pair}}$ and $y=\frac{TIT}{T_{g1}}$.Thermal efficiency
(2)$$\eta_{th_{GT}}=\frac{\left(1+far\right)\frac{c_{Pcg}}{c_{Pair}}y\eta_{t}\left(1-\frac{1}{\pi_{t}^{x_{cg}}}\right)-\frac{\pi_{C}^{x_{air}}-1}{\eta_{C}}}{\left(1+far\right)\frac{c_{Pcg}}{c_{Pair}}y-1-\frac{\pi_{C}^{x_{air}}-1}{\eta_{C}}}$$
where $x_{cg}=\frac{R_{cg}}{c_{Pcg}}$.Air mass flow rate used by the two simple gas turbines
(3)$${\dot{m}}_{air}=\frac{{\dot{W}}_{mGT}}{c_{Pair}T_{g1}\left[\left(1+far\right)\frac{c_{Pcg}}{c_{Pair}}y\eta_{t}\left(1-\frac{1}{\pi_{T}^{x_{cg}}}\right)-\frac{\pi_{C}^{x_{air}}-1}{\eta_{C}}\right]}$$In this equation, ${\dot{W}}_{mGT}$ is the power generated by the two gas turbines.Fuel mass flow rate fed to the pair of gas turbines
(4)$${\dot{m}}_{f}={\dot{m}}_{air}far$$Mass flow rate of combustion gases produced by the two Mitsubishi gas turbines
(5)$${\dot{m}}_{cg}={\dot{m}}_{air}\left(1+far\right)$$Power supplied to the compressors of both gas turbine cycles
(6)$${\dot{W}}_{C}={\dot{m}}_{air}\left(h_{g2}-h_{g1}\right)$$Exergy flow rates for the thermodynamic states of air and combustion gases are given respectively as follows:
(7a)$$\begin{array}{c} {\dot{E}}_{gi}={\dot{m}}_{air}\left[\left(h_{gi}-h_{0}\right)-T_{0}\left(s_{gi}-s_{0}\right)\right],\;\;\mathrm{for}\;i=1\;\mathrm{and}\;2 \end{array}$$
(7b)$$\begin{array}{c} {\dot{E}}_{gi}={\dot{m}}_{cg}\left[\left(h_{gi}-h_{0}\right)-T_{0}\left(s_{gi}-s_{0}\right)\right],\;\;\mathrm{for}\;i=3,\ldots ,13 \end{array}$$The thermal exergy flow rate provided by the combustion of the natural gas is
(8)$${\dot{E}}_{f}={\dot{m}}_{f}LHV\left(1-\frac{T_{0}}{T_{af}}\right)$$
where LHV and $T_{af}$ are respectively the low heating value and the adiabatic flame temperature of the natural gas.

The composition of the combustion gases is obtained from the following combustion reaction in the combustion chamber of the gas turbine
(9)$$\begin{array}{ll} C_{n}H_{m}+X_{stHA}\left(1+\lambda \right) & \left[X_{DA}\left(0.21O_{2}+0.79N_{2}\right)+X_{H_{2}O}H_{2}O\right]\\ & \to \alpha_{{\mathrm{CO}}_{2}}{\mathrm{CO}}_{2}+\alpha_{H_{2}O}H_{2}O+\alpha_{N_{2}}N_{2}+\alpha_{O_{2}}O_{2}+\alpha_{\mathrm{CO}}\mathrm{CO}+\alpha_{C_{n}H_{m}}C_{n}H_{m}+\alpha_{{\mathrm{NO}}_{x}}{\mathrm{NO}}_{x} \end{array}$$
where *n* and *m* are the number of carbons and hydrogens present in the natural gas; $X_{stHA}$, $X_{DA}$, and $X_{H_{2}O}$ are the stoichiometric coefficients of the humid air and the fractions of dry air and humid air, respectively; λ is the excess air, which is derived from the energy balance in the combustion chamber; $\alpha_{i}$ is the stoichiometric coefficient of the compounds present in the combustion reaction; and the coefficients $\alpha_{\mathrm{CO}}$, $\alpha_{C_{n}H_{m}}$, and $\alpha_{{\mathrm{NO}}_{x}}$ are computed from the Rizk and Mongia correlations [20].

### 2.3. Steam Cycle

In the steam cycle, steam flows to a steam turbine to generate mechanical energy, which is used to drive an electrical generator. The reduced-energy steam flows out of the turbine and enters the condenser, where it is condensed to the condition of saturated liquid. A feedwater pump returns the condensed liquid to the heat recovery steam generator. The steam cycle operates in agreement with the exergy–enthalpy diagram depicted in Figure 2b. Even if the streams of main and reheated steam, v1 and v4, have the same temperature, v1 presents the highest exergy content while the state v4 has the greatest energy content. The pressures associated with each of these states explain this fact since the pressure of v1 is greater than the pressure of v4. In addition, the exergy of the heat discarded in the condenser, $\varepsilon_{v6}-\varepsilon_{v7}$, is low, even if its energy content, $h_{v6}-h_{v7}$, is high because the condensation temperature is very close to the temperature of the surroundings (the dead state temperature). On the other hand, the exergy changes in the steam expansions, $\varepsilon_{v1}-\varepsilon_{v2}$ and $\varepsilon_{v4}-\varepsilon_{v6}$, are greater than their energy changes, $h_{v1}-h_{v2}$ and $h_{v4}-h_{v6}$, indicating that only a part of the steam exergy is used to generate work.

In the steam cycle, mass and energy conservation laws are applied to each component to analyze the thermodynamic performance of the steam cycle, and the following set of equations are obtained:Net work of the steam turbines
(10)$$\begin{array}{ll} w_{mSC}=h_{v1}-h_{v2} & +\left(1+\frac{{\dot{m}}_{IP}}{{\dot{m}}_{HP}}\right)\left(h_{v4}-h_{v5}\right)+\left(1+\frac{{\dot{m}}_{IP}}{{\dot{m}}_{HP}}+\frac{{\dot{m}}_{LP}}{{\dot{m}}_{HP}}\right)\left(h_{v5}-h_{v6}\right)\\ & -\left(h_{v14}-h_{v9b}\right)-\frac{{\dot{m}}_{IP}}{{\dot{m}}_{HP}}\left(h_{v11}-h_{v9a}\right)-\left(1+\frac{{\dot{m}}_{IP}}{{\dot{m}}_{HP}}+\frac{{\dot{m}}_{LP}}{{\dot{m}}_{HP}}\right)\left(h_{v8}-h_{v7}\right)-\frac{{\dot{W}}_{CWP}}{{\dot{m}}_{HP}} \end{array}$$Net power of the steam turbine
(11)$${\dot{W}}_{mSC}={\dot{m}}_{HP}w_{mSC}$$Recovered heat in the heat recovery steam generator
(12)$$\begin{array}{ll} q_{H_{HRSG}}=h_{v1}-h_{v14}+\frac{{\dot{m}}_{IP}}{{\dot{m}}_{HP}}\left(h_{v3a}-h_{v11}\right) & +\left(1+\frac{{\dot{m}}_{IP}}{{\dot{m}}_{HP}}\right)\left(h_{v4}-h_{v3}\right)\\ & +\frac{{\dot{m}}_{LP}}{{\dot{m}}_{HP}}\left(h_{v5a}-h_{v9c}\right)+\left(1+\frac{{\dot{m}}_{IP}}{{\dot{m}}_{HP}}+\frac{{\dot{m}}_{LP}}{{\dot{m}}_{HP}}\right)\left(h_{v9}-h_{v8}\right) \end{array}$$Thermal efficiency of the steam cycle
(13)$$\eta_{thST}=\frac{w_{mSC}}{q_{H_{HRSG}}}$$
The exergy flow rates of the thermodynamic states of steam are given by
(14)$${\dot{E}}_{vi}={\dot{m}}_{vi}\left[\left(h_{vi}-h_{0}\right)-T_{0}\left(s_{vi}-s_{0}\right)\right],\;\;\mathrm{for}\;i=1,\ldots ,17$$
The required mass flow rate of cooling water for the condenser is given by
(15)$${\dot{m}}_{CW}=\left({\dot{m}}_{HP}+{\dot{m}}_{IP}+{\dot{m}}_{LP}\right)\frac{\left(h_{v6}-h_{v7}\right)}{c_{PCW}\left(T_{CW3}-T_{CW2}\right)}$$
where $T_{CW2}$ is the cooling water at the exit of the cooling system pump and $T_{CW3}$ is the temperature of the cooling water at the exit of the condenser, where $T_{CW3}=T_{v6}-T_{COND}$ with $\Delta T_{COND}$ being the heat exchange temperature difference in the condenser.

The power used by the pump circulating the cooling fluid through the condenser is given by
(16)$${\dot{W}}_{CWP}=\eta_{CWP}{\dot{m}}_{CW}c_{PCW}\left(T_{CW2}-T_{CW1}\right)$$
where $T_{CW1}$ is the cooling water at the entrance of the cooling system pump.

Finally, from the definition of the exergy transfer accompanying a heat flow, the exergy of the waste heat dissipated in the condenser results given by
(17)$${\dot{E}}_{{\dot{Q}}_{COND}}=\left({\dot{m}}_{HP}+{\dot{m}}_{IP}+{\dot{m}}_{LP}\right)\left(h_{v6}-h_{v7}\right)\left(1-\frac{T_{0}}{T_{\mathrm{COND}}}\right)$$

In the above equation $q_{COND}=h_{v7}-h_{v6}$ and $T_{\mathrm{COND}}$ are the condensation enthalpy and the saturation temperature of water respectively, both at $P_{COND}$.

### 2.4. Heat Recovery Steam Generator (HRSG)

The HRSG has three pressure levels: low pressure, intermediate pressure, and high pressure. Each pressure level includes three main groups of heat exchangers: economizer, evaporator, and superheater. When the gas turbine exhaust gases pass over the HRSG heating elements, the water inside the tubes recovers (absorbs) energy from the hot exhaust gases and changes its phase into steam. The produced steam is used to drive steam turbines and to generate shaft power in a steam cycle. Water preheating and evaporation occur in economizers and evaporators, respectively. After separating the liquid water and steam in the drum, water goes through the evaporator while steam enters the superheater, as shown in Figure 1.

An important design parameter for analyzing the HRSG is the pinch-point temperature difference. This is the difference between the temperature of the gas turbine exhaust exiting the evaporator and the temperature of water evaporation. As shown in Figure 1, the evaporators pinch-point temperature differences are then given by the following:
(18a)$$\begin{array}{c} \Delta T_{pp\;HP}=T_{g6}-T_{v16},\;T_{v16}=T_{sat}\left(P_{HP}\right) \end{array}$$
(18b)$$\begin{array}{c} \Delta T_{pp\;IP}=T_{g10}-T_{v12},\;T_{v12}=T_{sat}\left(P_{IP}\right) \end{array}$$
(18c)$$\begin{array}{c} \Delta T_{pp\;LP}=T_{g12}-T_{v9},\;T_{v9}=T_{sat}\left(P_{LP}\right) \end{array}$$
The combustion gase temperatures are calculated from the energy balance for gas and water in each heating element of the HRSG. The energy balances of the triple-pressure HRSG with preheating and reheating are given by the following:High pressure superheater and intermediate pressure reheater (HPSH + IPRH)
(19)$${\dot{m}}_{cg}c_{Pcg\left(g4-g5\right)}\left(T_{g4}-T_{g5}\right)={\dot{m}}_{HP}\left(h_{v1}-h_{v17}\right)+\left({\dot{m}}_{HP}+{\dot{m}}_{IP}\right)\left(h_{v4}-h_{v3}\right)$$High pressure evaporator (HPEV)
(20)$${\dot{m}}_{cg}c_{Pcg\left(g5-g6\right)}\left(T_{g5}-T_{g6}\right)={\dot{m}}_{HP}\left(h_{v17}-h_{v16}\right)$$High pressure economizer (HPEC)
(21)$${\dot{m}}_{cg}c_{Pcg\left(g6-g7\right)}\left(T_{g6}-T_{g7}\right)={\dot{m}}_{HP}\left(h_{v16}-h_{v15}\right)$$Low pressure superheater (LPSH)
(22)$${\dot{m}}_{cg}c_{Pcg\left(g7-g8\right)}\left(T_{g7}-T_{g8}\right)={\dot{m}}_{LP}\left(h_{v5a}-h_{v10}\right)$$Intermediate pressure superheater (IPSH)
(23)$${\dot{m}}_{cg}c_{Pcg\left(g8-g9\right)}\left(T_{g8}-T_{g9}\right)={\dot{m}}_{IP}\left(h_{v3a}-h_{v13}\right)$$Intermediate pressure evaporator (IPEV)
(24)$${\dot{m}}_{cg}c_{Pcg\left(g9-g10\right)}\left(T_{g9}-T_{g10}\right)={\dot{m}}_{IP}\left(h_{v13}-h_{v12}\right)$$High pressure preheater and Intermediate pressure economizer (HPPH+IPEC)
(25)$${\dot{m}}_{cg}c_{Pcg\left(g10-g11\right)}\left(T_{g10}-T_{g11}\right)={\dot{m}}_{HP}\left(h_{v15}-h_{v14}\right)+{\dot{m}}_{IP}\left(h_{v12}-h_{v11}\right)$$Low pressure evaporator (LPEV)
(26)$${\dot{m}}_{cg}c_{Pcg\left(g11-g12\right)}\left(T_{g11}-T_{g12}\right)={\dot{m}}_{LP}\left(h_{v10}-h_{v9c}\right)$$Low pressure economizer (LPEC)
(27)$${\dot{m}}_{cg}c_{Pcg\left(g12-g13\right)}\left(T_{g12}-T_{g13}\right)=\left({\dot{m}}_{HP}+{\dot{m}}_{IP}+{\dot{m}}_{LP}\right)\left(h_{v9}-h_{v8}\right)$$

In Equations (Equation 19)–(Equation 27), $c_{Pcg}=\sum_{i}\frac{\alpha_{i}}{\alpha_{tot}}c_{Pi}\left(T_{av}\right)$, where $T_{av}$ is the average temperature of the combustion gase temperature at the entrance and exit of each section of the HRSG and $\alpha_{tot}=\sum_{i}\alpha_{i}$ for i=CO${}_{2}$, H${}_{2}$O, N${}_{2}$, O${}_{2}$, CO, NO${}_{x}$, and C${}_{n}$H${}_{m}$. The specific heat capacity of each compound, $c_{Pi}$, is determined by using the Rivkin correlations [21].

### 2.5. Performance Parameters of the Combined Cycle

The combinations of energy and mass balance equations are numerically solved to get the temperature profile of the gas and water/steam side of HRSG. The combined cycle performance parameters considered in this work are as follows:The specific fuel consumption of the combined cycle is given by
(28)$$SFC_{CC}=\frac{3600{\dot{m}}_{f}}{{\dot{m}}_{air}c_{Pair}T_{g1}\left[\left(1+far\right)\frac{c_{Pcg}}{c_{Pair}}y\eta_{t}\left(1-\frac{1}{\pi_{t}^{x_{cg}}}\right)-\frac{\pi_{C}^{x_{air}}-1}{\eta_{C}}\right]+{\dot{W}}_{mSC}}$$The thermal efficiency of the combined cycle is determined by the following expression
(29)$$\eta_{thCC}=\frac{{\dot{W}}_{mGT}+{\dot{W}}_{mSC}}{{\dot{m}}_{air}c_{Pair}T_{g1}\left[\left(1+far\right)\frac{c_{Pcg}}{c_{Pair}}y-1-\frac{\pi_{C}^{x_{air}}-1}{\eta_{C}}\right]}$$The specific steam consumption of the combined cycle is given by
(30)$$SSC_{CC}=\frac{3600{\dot{m}}_{HP}}{{\dot{m}}_{air}c_{Pair}T_{g1}\left[\left(1+far\right)\frac{c_{Pcg}}{c_{Pair}}y\eta_{t}\left(1-\frac{1}{\pi_{t}^{x_{cg}}}\right)-\frac{\pi_{C}^{x_{air}}-1}{\eta_{C}}\right]+{\dot{W}}_{mSC}}$$

## 3. Exergetic Cost Analysis

### 3.1. Productive Structure

The productive structure of the three-pressure-level combined cycle is presented in Figure 3. A productive structure is a graphical representation of the exergy flow interactions of the plant components on the basis of their productive objective. The inputs of each component are the resources $\left(\dot{F}\right)$, and the outputs are the products $\left(\dot{P}\right)$. The productive structure also helps to identify the distribution of the resources and internal products throughout the plant, using the physical model as reference [2]. As shown in Table 2 and Figure 3, the external resources of the combined cycle of three pressure levels are the air, fuel, and water; the product is the mechanical energy produced by the gas turbine cycle and the steam cycle; and the residual streams are the exhaust gases and the heat dissipated in the condenser.

The components of the productive structure are the same as those of the physical structure, and they are connected by lines corresponding to exergy flows. Among these components, there are productive and dissipative components.

#### 3.1.1. Productive Components

The purpose of the productive components is to provide resources to the other components, and they are all involved in the formation processes of the products and residues of an energy system. According to the productive objective of each component, the resources $\left(\dot{F}\right)$, products $\left(\dot{P}\right)$, and irreversibilities $\left(\dot{I}\right)$ of each productive component of the combined cycle and the global plant are summarized in Table 2. These exergy flows are related by the exergy balance given by the following expression
(31)$${\dot{F}}_{i}-{\dot{P}}_{i}-{\dot{I}}_{i}=0$$

The exergetic efficiency of a productive component is defined as the ratio between its product and resource exergy flows:(32)$$\eta_{ex}=\frac{\mathrm{Product}\;\mathrm{Exergy}\;\mathrm{flow}\;\mathrm{rate}}{\mathrm{Resource}\;\mathrm{flow}\;\mathrm{rate}\;\mathrm{fuel}}=\frac{\dot{P}}{\dot{F}}$$

The system studied in this work is composed of three main subsystems, the gas turbine cycle (GT), the heat recovery steam generator (HRSG), and the steam cycle (SC). The productive components of the gas turbine are the compressor (C), the combustion chamber (cc), and the expansion turbine (t). The productive purpose of the compressor is to increase the exergy of the air by increasing its pressure (mechanical energy) using the compression power as fuel. The combustion chamber increases the air temperature through the exothermic reaction between air and natural gas. The resource of this productive component is, therefore, the exergy of natural gas combustion heat. The product of the expansion turbine is the generated power to mechanically drive the compressor and for external use, using the difference between exiting and entering combustion gases exergies as resource. In the combined cycle framework, the resources of the gas turbine subsystem are the exergy flows of air and natural gas and their products are the output net power of the gas turbine and the exergy flow of the combustion gases leaving the expansion turbine. It is important to notice that, in this context, the gas turbine has no residues, since the exergy flow of the combustion gases, ${\dot{E}}_{g4}$, is the resource of the HRSG.

The HRSG is the connection between the gas turbine and steam cycles: the high-, intermediate-, and low-pressure steam obtained from the HRSG is used by steam turbine (ST). The productive components of the HRSG are the high-pressure superheater independently coupled with intermediate the pressure reheater (HPSH + IPRH); the high-, intermediate-, and low-pressure evaporators (HPEV, IPEV, and LPEV); the high-, intermediate-, and low-pressure economizers (HPEC, IPEC, and LPEC); the high-, intermediate-, and low-pressure superheaters (HPSH, IPSH, and LPSH); and the high pressure preheater independently coupled with the intermediate pressure economizer (HPPH + IPEC). The resources of these heat exchangers are the exergy differences between exiting and entering combustion gases, while their products are exergy differences between exiting and entering water. In this way, the productive objective of this set of heat exchangers is to increase the exergy of water; by increasing its thermal energy, at the same time, they save part of the exergy of the combustion generated in the gas turbine cycle.

The productive components of the steam cycle are the high-, intermediate-, and low-pressure steam turbines (HPST, IPST, and LPST); the low-, intermediate-, and high-pressure pumps (LPP, IPP, and HPP); the low-, intermediate-, and high-pressure drums (LPD, IPD, and HPD); and the mixer (M), which mixes the streams of water leaving the HPST and the IPSH to produce the stream of water entering the IPRH. The resources of the HPST and IPST are the exergy differences between the entering and exiting superheated steam at high and intermediate pressure. The resource of the LPST is the exergy difference between the entering superheated steam at low pressure and the leaving wet steam at $P_{COND}$. The product of the three steam turbines is the mechanical power used to drive the electric generator. The resource of the LPP, IPP, and HPP is the power consumption of the pumps; their products are the exergy differences between exiting and entering water; and their productive objective is to increase the pressure of water. Even if the productive purpose of the LPD, IPD, and HPD is to separate satured liquid water from saturated steam, in this work, they are treated only as nodes. The irreversibilities associated to these three components take all the zero value.

In summary, the set of productive components $\left(P\right)$ is conformed by 24 components and is the union of the components of the gas turbine cycle (3 components: C, cc, and t), the heat recovery steam generator (9 components: HPSH+IPRH, HPEV, HPEC, LPSH, IPSH, IPEV, HPPH+IPEC, LPEV, and LPEC), and the steam cycle without the condenser (12 components: HPST, IPST, LPST, LPP, IPP, HPP, LPD, IPD, HPD, m, M, and EG), $P=GT\cup HRSG\cup \left(SC-\left\{COND\right\}\right)$.

#### 3.1.2. Dissipative Components

Every energy system has residual streams without any utility and that formed along the formation process of the products of an energy system. The exergy of these streams can be constituted by different types of exergy depending on their intensive potentials (temperature, pressure, concentration, and velocity) of the stream with respect to the dead state. The residue exergy of a residual stream can be destroyed in one or several components, known as dissipative components, with or without gaining something thermodynamically useful directly from the same components. The exergy of a residual stream may therefore result in more than one residue, and each and every residue leaves the plant but needs additional fuel to get rid of it. The dissipative components are responsible for discarding the residues to the environment, and according to Lazzaretto et al., they are components in which exergy is destroyed without gaining something thermodynamically useful directly from the same component [22]. Even if they have not a clear productive purpose, they are necessary for the plant operation, and their operation is only significant when they are considered in the context of the overall system.

The dissipative components serve the productive components in a system (they help to reduce the destruction of exergy in at least one of the components remaining of the system), assist in reducing the investment costs of the entire system, or allow the system to fulfill the required emission standards [22]. For this components, the exergetic efficiency is meaningless, unless it is considered together with the components it serves. In the case of the paper, the combined cycle plant has two residual streams:The exergy flow rate of the exhaust gases releases to the environment through the stack. Strictly, this stream contains at least three residues: its physical exergy, the fraction of its chemical exergy associated to CO${}_{2}$, and that associated to NOx. However, since in this work the chemical exergy of the working fluids is neglected, the residual stream of exhausted gases corresponds to their physical exergy, which also coincides with the residue associated to the formation of the power generated by the gas turbines, ${\dot{R}}_{stack}={\dot{E}}_{g13}$.It should be appointed that the HRSG, as it can be observed in Figure 2a, saves only a part of the exergy generated in the productive components of the gas turbine (g4 to g13) and that the stack together with the environment close the gas turbine cycle by reducing in the atmosphere the chemical and physical exergy of the exhaust gases to reach the dead state (g13 to 0). On the other hand, the coupling of the gas turbine and the steam cycle through the HRSG transfers the residue associated to the combustion gases from the gas turbine to the HRSG, since they are produced by the gas turbine and used as a resource by the HRSG.The exergy of wet steam leaves the low-pressure stream turbine, corresponding to state v6. As shown in Table 3, ${\dot{E}}_{v6}$ is the resource of the condenser, of which the productive purpose is to close the thermodynamic cycles of steam or, in other words, to destroy the exergy of wet steam by its condensation from state v6 to v7. This is made by the heat exchange from steam to cooling water using the power of cooling water pump as additional resource. If the cooling water is assumed to be the dead state for water, then the residue of the condenser is ${\dot{R}}_{COND}={\dot{E}}_{{\dot{Q}}_{COND}}$.

Therefore, the residues of the combined cycle plant are the exhaust gases (g13) delivered to the environment from a dissipative component usually called stack and the waste heat dissipated from the condenser. The set of dissipative components (D) responsible for releasing the residues of combined cycle to the environment is therefore D=stack,COND.

### 3.2. The Exergetic Costs Model

#### 3.2.1. The Cost Formation Process of Residues

The cost formation process is the process through which the cost of the consumed resources are gradually charged to the material streams, increasing their exergetic cost when passing from the beginning to the end of the “productive chain”m and at the same time, exergy is gradually destroyed [23,24]. In this way, as there is a process of cost formation of the functional products, there also exists a cost formation process of the residues.

The residue exergetic cost of the entire power plant is the sum of the exergetic cost of the residues released in all the dissipative components $\left(R_{r}^{*}\right)$:(33)$$R^{*}=\sum_{r\in D}R_{r}^{*}$$
If the residue dissipated to the environment in the *r*th dissipative component $\left(r\in D\right)$ has its origin in several components, the cost of this residue $R_{r}^{*}$ is decomposed into several costs, one for each component that has generated it:(34)$$R_{r}^{*}=\sum_{i\in P}R_{ri}^{*},\;r\in D$$
where P is the set of all the productive components forming the residue dissipated in the *r*th dissipative component and $R_{ri}^{*}$ is the exergy cost of the residue dissipated in the *r*th component that has been generated by the *i*th productive component. The cost of the residues charged to the *i*th productive component is then given by
(35)$$R_{i}^{*}=\sum_{r\in D}R_{ri}^{*},\;i\in P$$
To determine the values of $R_{ri}^{*}$ for r∈D, the exergoeconomic theory defines the residues cost allocation ratios. For a given residue dissipated in the *r*th dissipative component, the residue cost allocation ratio associated to the *i*th productive component $\left(\rho_{ri}\right)$ is the fraction of the exergetic cost of the residue which is allocated to this component in such a way that
(36)$$R_{ri}^{*}=\rho_{ri}R_{r}^{*},\;\;\mathrm{with}\sum_{i\in P}\rho_{ri}=1,\;r\in D$$

The residue cost allocation ratios determine how the cost of the residue that leaves the system should be decomposed into several costs. Nevertheless, as shown in Section 3.3, there is no definitive way to determine these ratios.

According to Section 3.1.2, the residue of the plant is given by $R^{*}=R_{stack}^{*}+R_{COND}^{*}$, and since $R_{stack}^{*}={\dot{E}}_{g13}^{*}$ and $R_{COND}^{*}={\dot{E}}_{{\dot{Q}}_{COND}}^{*}$, then
(37)$$R^{*}=R_{stack}^{*}+R_{COND}^{*}={\dot{E}}_{g13}^{*}+{\dot{E}}_{{\dot{Q}}_{COND}}^{*}$$
From Equations (Equation 35) and (Equation 36), it results that the cost of the residues charged to the *i*th productive component is then given by
(38)$$R_{i}^{*}=R_{stack,i}^{*}+R_{COND,i}^{*}=\mu_{i}{\dot{E}}_{g13}^{*}+\beta_{i}{\dot{E}}_{{\dot{Q}}_{COND}}^{*},\;i\in P,\;\mathrm{with}\sum_{i\in P}\mu_{i}=1=\sum_{i\in P}\beta_{i}$$
where $\mu_{i}=\rho_{stack,i}$ and $\beta_{i}=\rho_{COND,i}$ are respectively the exhaust gases and the waste heat dissipated from the condenser cost allocation ratios for the *i*th productive component. It should be noted that, if a productive component does not participate in the formation of a given residue, then its corresponding residue cost allocation ratio vanishes.

#### 3.2.2. Exergetic Costs Equations

The exergetic costs of the streams are obtained from the solution of the cost balances of the components and auxiliary equations, which are derived from the exergy costing rules, also known as, resource–product propositions [2,8]:Exergetic cost equations for the external resourcesThe costs of the external resources are known
(39)$$E_{f}^{*}={\dot{E}}_{f},\;E_{g1}^{*}={\dot{E}}_{g1}=0\;\mathrm{and}\;E_{v7}^{*}={\dot{E}}_{v7}$$Exergetic cost balance equation for a productive componentAccording to the conservative nature of costs, the product cost of the component *i*$\left(P_{i}^{*}\right)$ is equal to the cost of resources required to obtain it $\left(F_{i}^{*}\right)$ plus the cost of the residues allocated to the *i*th component $\left(R_{i}^{*}\right)$.
(40)$$\begin{array}{cc} \; & \;\;\;\;\left.\begin{array}{c} P_{i}^{*}=F_{i}^{*}+R_{i}^{*}\Rightarrow P_{i}^{*}=F_{i}^{*}+\mu_{i}{\dot{E}}_{g13}^{*}+\beta_{i}{\dot{E}}_{{\dot{Q}}_{COND}}^{*},\;i\in P\\ F_{i}^{*}={\dot{F}}_{e\to i}+\sum_{j}{\dot{F}}_{j\to i} \end{array}\right. \end{array}$$The cost of the resources consumed by the component *i* is composed by the cost of external resources $\left({\dot{F}}_{e\to i}\right)$ and the cost of flows coming from other productive component *j*$\left({\dot{F}}_{j\to i}\right)$.Exergetic cost balance equation for each dissipative componentThe exergy cost balance equation for the *r*th dissipative component, in which the residue ${\dot{R}}_{r}$ is released to the environment, is given by
(41)$$R_{r}^{*}=F_{P,r}^{*}+F_{e,r}^{*},\;r\in D=\left\{stack,COND\right\}$$
where $F_{P,r}^{*}$ and $F_{e,r}^{*}$ are respectively the formation and elimination costs of the residue. For the residues considered in this work, $R_{stack}={\dot{E}}_{g13}$ and $R_{COND}={\dot{E}}_{{\dot{Q}}_{COND}}$, the formation costs are $F_{P,stack}^{*}=E_{g13}^{*}$ and $F_{P,COND}^{*}=E_{v6}^{*}-E_{v7}^{*}$ while the elimination costs are $F_{e,stack}^{*}=0$ and $F_{e,COND}^{*}={\dot{W}}_{CWP}$.Auxiliary equations of components with multiple productsIf a unit has a product composed of several flows, then the same unit exergetic cost can be assigned to them. In fact, even if two or more products can be identified in the same unit, their formation processes are inseparable at the level of aggregation considered, and therefore, a cost proportional to their exergy is assigned to them.Cost balance equations for the complete combined cycleThe cost balance equation for the entire power plant [25] is given by
(42)$$F_{CC}^{*}=P_{u,CC}^{*}\Rightarrow E_{g1}^{*}+E_{f}^{*}+E_{v7}^{*}=W_{mCC}^{*}$$
where $F_{CC}^{*}$ and $P_{u,CC}^{*}$ are respectively the exergetic costs of the external resources and the useful product of the combined cycle.

For this study, the set of cost equations, presented in Table 4, conform a system of linear equations for the n=48 exergetic costs of the material and energy streams of the power plant, denoted as $E^{*}$. The components of the square coefficient matrix of the system, $A\in M_{n\times n}$, also known as the incidence matrix, is a function of the exergy flow rates of the streams and the residues cost allocation ratios β and μ [26]. In matrix notation, the systems of equations can be expressed as
(43)$$AE^{*}=F_{e}^{*}$$
where $F_{e}^{*}\in M_{n\times 1}$ is the vector of exergetic costs of the external resources.

### 3.3. Criteria for Residues Cost Allocation

The calculation of the production costs including the residue formation depends on their allocation to productive components. In this work, the residue cost allocation is based on the premise that residues must be allocated to the productive components that generated their costs, and it is done by means of the so-called residue cost allocation ratios. However, there is no a definitive way to determine these ratios, and several criteria have been proposed in the literature. This fact motivates to propose a criterion based on the irreversibilities generated along with the residue formation process. In this section, four different criteria for residue cost allocation are presented:**C1:** Generated entropy along the process [2,6,27],**C2:** Generated irreversibilities along the process (proposed in this work),**C3:** Distributed exergy along the process [2,3,16] and**C4:** Distributed entropy along the process [16,17,28].

#### 3.3.1. Criterion of Entropy Changes in the Productive Components (C1)

The air and natural gas are the external resources of the gas turbine, which operates following an open Brayton cycle. As it can be appreciated from Figure 4a, the total entropy change in the processes occurring within the components of the gas turbine is positive:(44)$$\Delta {\dot{S}}_{GT}=\sum_{k\in GT}\Delta {\dot{S}}_{k,g}=\left({\dot{S}}_{g1}-{\dot{S}}_{g4}\right)>0$$
where $\Delta {\dot{S}}_{k,g}$ is the entropy change of air or combustion gases in the *k*th component of the gas turbine. Each entropy change is obtained from the expression of the entropy change for an ideal gas considering that the heat capacity at constant pressure remains constant, ${\dot{S}}_{g1}-{\dot{S}}_{a0}={\dot{m}}_{air}\left[c_{Pair}\ln\left(T_{g1}/T_{0}\right)-R_{air}\ln\left(P_{g1}/P_{0}\right)\right]$ and ${\dot{S}}_{g4}-{\dot{S}}_{cg0}={\dot{m}}_{cg}\left[c_{Pcg}\ln\left(T_{g4}/T_{0}\right)-R_{cg}\ln\left(P_{g4}/P_{0}\right)\right]$.

In the combined cycle, the thermal energy of the combustion gases leaving the gas turbine is exploited in the components of the HRSG to generate steam. As they pass through the different sections of the HRSG, the entropy of the combustion gases decreases, as it can be observed from Figure 4a, and thus, the total entropy change of the combustion gases in the HRSG is negative:(45)$$\Delta {\dot{S}}_{HRSG,g}=\sum_{k\in HRSG}\Delta {\dot{S}}_{k,g}={\dot{S}}_{g13}-{\dot{S}}_{g4}<0$$

The HRSG helps to compensate a fraction of the entropy increase in the gas turbine, and since in the stack there is no entropy change of the exhaust gases ($\Delta {\dot{S}}_{stack}=0$), then the remaining fraction $\Delta {\dot{S}}_{ENV}={\dot{S}}_{cg0}-{\dot{S}}_{g13}<0$ is compensanted by the cooling of the exhaust gases in the environment. In this approach, the purpose of the environment, which is a dissipative component, is to decrease the entropy of the combustion gases in such a way that the open Brayton cycle becomes a closed cycle (see the red line of Figure 4a), and therefore
(46)$$\sum_{k\in \Omega_{g}}\Delta {\dot{S}}_{gk}+\left({\dot{S}}_{cg0}-{\dot{S}}_{g13}\right)=\underset{>0}{\underset{︸}{\Delta {\dot{S}}_{GT}}}+\underset{<0}{\underset{︸}{\Delta {\dot{S}}_{HRSG,g}}}+\underset{=0}{\underset{︸}{\Delta {\dot{S}}_{stack}}}+\underset{<0}{\underset{︸}{\Delta {\dot{S}}_{ENV}}}=0$$
where $\Omega_{g}=GT\cup HRSG$ is the set of productive components of the gas turbine and HRSG involved in the entropy changes of air and combustion gases to form the residual stream g13.

In criterion C1, the formation cost of the exhaust gases is allocated to the productive components of the combined cycle involved in their formation process in proportion to the ratio between the entropy change in a component and the sum of the total entropy changes in all the productive components participating in its formation process $\left(\Delta {\dot{S}}_{tot,g}^{prod}=\Delta {\dot{S}}_{GT}+\Delta {\dot{S}}_{HRSG,g}\right)$. In this way, the expression of the exhaust gase cost allocation ratio for the *k*th productive component of $\Omega_{g}$, $\mu_{k}^{\dot{S}}$, is given by
(47)$$\mu_{k}^{\dot{S}}=\frac{\Delta {\dot{S}}_{k,g}}{\Delta {\dot{S}}_{tot,g}^{prod}}=-\frac{\Delta {\dot{S}}_{k,g}}{\Delta {\dot{S}}_{ENV}},\;\;k\in \Omega_{g}=GT\cup HRSG\;\mathrm{and}\;\;\sum_{k\in \Omega_{g}}\mu_{k}^{\dot{S}}=1$$

The coefficients $\mu_{k}^{\dot{S}}$ represent the fraction of the formation cost of the exhaust gases allocated to the *k*th productive component. For the compressor and the expansion turbine, $0<\mu_{k}^{\dot{S}}<1$; for the combustion chamber, $\mu_{k}^{\dot{S}}>1$; for the components of the HRSG, $\mu_{k}<0$; and their summation is equal to one. This means that the components with values of $\mu_{k}^{\dot{S}}$ greater than 0 contribute to the formation cost of the exhaust gases and those of which values are lesser than zero receive a credit. For its part, the combustion chamber is the component that generates the greatest amount of entropy, and therefore, it is the more penalized productive component.

As it can be observed from Figure 4b together with Figure 1, the steam cycle is a closed cycle. The entropy increases of steam occur in all the productive components of the HRSG and the steam cycle $\left(\Omega_{v}=HRSG\cup SC_{prod}\right)$, and they equal in magnitude the entropy saved in the condenser to reach saturated liquid condition (see red line of Figure 4b),
(48a)$$\begin{array}{cc} \Delta {\dot{S}}_{tot,v}^{prod}=\Delta & {\dot{S}}_{HRSG,v}+\Delta {\dot{S}}_{SC,prod}>0 \end{array}$$
(48b)$$\begin{array}{cc} \; & \Delta {\dot{S}}_{COND}<0 \end{array}$$
(48c)$$\begin{array}{cc} \Delta {\dot{S}}_{tot,v}^{prod} & +\Delta {\dot{S}}_{COND}=0 \end{array}$$
where
(49a)$$\begin{array}{cc} \Delta {\dot{S}}_{HRSG,v}={\dot{S}}_{v1}+{\dot{S}}_{v4}+{\dot{S}}_{v5a}+{\dot{S}}_{v3a}+{\dot{S}}_{v9} & -{\dot{S}}_{v3}-{\dot{S}}_{v9c}-{\dot{S}}_{v14}-{\dot{S}}_{v11}-{\dot{S}}_{v8} \end{array}$$
(49b)$$\begin{array}{cc} \Delta {\dot{S}}_{COND}= & {\dot{S}}_{v7}-{\dot{S}}_{v6} \end{array}$$
(49c)$$\begin{array}{cc} \Delta {\dot{S}}_{SC,prod}=-\Delta & {\dot{S}}_{HRSG,v}-\Delta {\dot{S}}_{COND} \end{array}$$

The cost of the waste heat dissipated from the condenser is allocated to the components of the HRSG and the productive components of the steam cycle ($SC_{prod}$) according to
(50)$$\beta_{k}^{\dot{S}}=\frac{\Delta {\dot{S}}_{k,v}}{\Delta {\dot{S}}_{tot,v}^{prod}}=-\frac{\Delta {\dot{S}}_{k,v}}{\Delta {\dot{S}}_{COND}},\;\;k\in \Omega_{v}=HRSG\cup SC_{prod}\;\mathrm{and}\;\;\sum_{k\in \Omega_{v}}\beta_{k}^{\dot{S}}=1$$

The values of $\beta_{k}^{\dot{S}}$ are between 0 and 1 for all the components of $\Omega_{v}$, and therefore, all these components contribute to the cost of this residue.

This criterion has been used by Ye and Li [11] and is equivalent to the criterion frequently used to allocate the cost of residues proportionally to entropy or negentropy generation along the processes of productive components [2,3,6,16,27,29]. In this approach, the productive purpose of the dissipative components of a system is to generate negentropy to compensate the entropy generation in the productive components. In this way, the environment produces the negentropy necessary to bring the exhaust gases into equilibrium with the ambient, the condenser also generates negentropy by decreasing the steam entropy to reach the saturated liquid condition, and the HRSG generates negentropy for the combustion gases and entropy for the steam [29].

It should be appointed that, in general, the entropy change and the entropy generation in a process are different because, according to the second law of thermodynamics [30], the total entropy generation in a process $\left({\dot{S}}_{gen,tot}\right)$ is given by
(51)$${\dot{S}}_{gen,tot}={\dot{S}}_{gen,CV}+{\dot{S}}_{gen,surr}=\frac{dS_{CV}}{dt}+\sum_{\mathrm{out}}{\dot{S}}_{e}-\sum_{\mathrm{in}}{\dot{S}}_{i}-\frac{{\dot{Q}}_{0}}{T_{0}}-\sum_{r}\frac{{\dot{Q}}_{r}}{T_{r}}$$
where ${\dot{Q}}_{r}$ is the rate of heat transfer from the *r*th thermal energy reservoir to the process volume control and $T_{r}$ is the temperatures of *r*th thermal energy reservoir. Equation (Equation 51) shows that the entropy change and the entropy generation are equal only if the control volume operates at steady-state and adiabatically.

#### 3.3.2. Criterion of Irreversibilities (C2)

The Gouy–Stodola theorem of thermodynamics states that the rate of work lost due to internal and external irreversibilities in the process is equal, in one hand, to the product of the dead state temperature and the total rate of entropy generation in the process and, on the other hand, to the total rate (internal and external) of exergy destruction in the process, which in thermoeconomics is also known as the rate of irreversibilities generated in the process,
(52)$${\dot{W}}_{lost}=T_{0}{\dot{S}}_{gen,tot}={\dot{E}}_{D,tot}=\dot{I}$$

The irreversibility of each component of the combined cycle is presented in Table 2. The criterion of irreversibilities is based on the accounting of irreversibilities throughout the formation process of each residue. For the combined cycle, the exhaust gases formation is accompanied by the irreversibilities generation in the processes of the gas turbine and the HRSG on the gas side to produce power and heat while the heat discarded in the condenser is the result of the irreversibilities generated in the generation processes of steam generation in the HRSG and power in the productive components of the steam cycle. However, in this case, this criterion faces the coupling of the generation of irreversibilities on the gas and steam sides within the HRSG. To deal with this problem, the state of aggregation in the HRSG is increased by identifying that each of its component can be understood as two subsystems (one for gases and one for steam) exchanging heat, as shown in Figure 5a.

The irreversibilities generated in the gas turbines are given by
(53)$${\dot{I}}_{GT}=\sum_{k\in GT}{\dot{I}}_{k}=T_{0}\left(\Delta {\dot{S}}_{GT}-\frac{{\dot{m}}_{f}LHV}{T_{af}}\right)$$

The integration of the gas turbines and the steam cycle takes places in the HRSG. In this work, the components of the HRSG are treated as adiabatic heat exchangers, in which, the thermal energy of the combustion gases is used to the steam generation.

The components of the HRSG are conceived as a subsystem conformed by two thermal energy reservoirs exchanging heat $\left({\dot{Q}}_{k}\right)$, as shown in Figure 5, one corresponding to the combustion gases and the other to the steam. For the combustion gases, ${\dot{E}}_{{\dot{Q}}_{k}}$ is the product, while for the steam reservoir, this heat exergy is the resource. In this context, the irreversibility of each component of the HRSG results as
(54)$${\dot{I}}_{HRSG,k}={\dot{F}}_{HRSG,k}-{\dot{E}}_{{\dot{Q}}_{k}}+{\dot{E}}_{{\dot{Q}}_{k}}-{\dot{P}}_{HRSG,k}=T_{0}\left(\Delta {\dot{S}}_{HRSG,k}^{g}+\frac{{\dot{Q}}_{k}}{{\langle T\rangle }_{k}^{g}}\right)+T_{0}\left(\Delta {\dot{S}}_{HRSG,k}^{v}-\frac{{\dot{Q}}_{k}}{{\langle T\rangle }_{k}^{g}}\right)$$
where ${\dot{F}}_{HRSG,k}=\Delta {\dot{E}}_{HRSG,k}^{g}$ and ${\dot{P}}_{HRSG,k}=\Delta {\dot{E}}_{HRSG,k}^{v}$ are respectively the resource and the product of the *k*th component of the HRSG. ${\dot{E}}_{{\dot{Q}}_{k}}$ is the maximum available heat transfer from the combustion gases within the *k*th HRSG component (see Figure 6),
(55)$${\dot{E}}_{{\dot{Q}}_{k}}=\left|{\dot{Q}}_{k}\right|\left(1-\frac{T_{0}}{{\langle T\rangle }_{k}^{g}}\right)$$
where ${\dot{Q}}_{k}=|\Delta {\dot{H}}_{HRSG,k}^{g}|=|\Delta {\dot{H}}_{HRSG,k}^{v}|$ and where ${\langle T\rangle }_{k}^{g}=2T_{k,in^{g}}T_{k,out}^{g}/\left(T_{k,in}^{g}+T_{k,out}^{g}\right)$ is an average temperature with $T_{k,in}^{g}$ and $T_{k,out}^{g}$ being the temperatures of the exhaust gases at the entrance and exit of the *k*th component of the HRSG respectively (see Figure 5 and Appendix A).

The irreversibility of the HRSG is the sum of the irreversibilities of all their components,
(56)$$\begin{array}{cc} {\dot{I}}_{HRSG}= & {\dot{F}}_{HRSG}-\sum_{k\in HRSG}{\dot{E}}_{{\dot{Q}}_{k}}+\sum_{k\in HRSG}{\dot{E}}_{{\dot{Q}}_{k}}-{\dot{P}}_{HRSG,k}\\ = & T_{0}\left(\Delta {\dot{S}}_{HRSG,g}+\sum_{k\in HRSG}\frac{{\dot{Q}}_{k}}{{\langle T\rangle }_{k}^{g}}\right)+T_{0}\left(\Delta {\dot{S}}_{HRSG,v}-\sum_{k\in HRSG}\frac{{\dot{Q}}_{k}}{{\langle T\rangle }_{k}^{g}}\right) \end{array}$$

The steam generated in the HRSG is used in the steam cycle to produce more power to the detriment of dissipating heat in the condenser. The irreversibilities generated along the processing of steam in the productive components of the steam cycle are given by
(57)$${\dot{I}}_{SC,prod}=\sum_{k\in SC,prod}{\dot{I}}_{k}=-T_{0}\left(\Delta {\dot{S}}_{HRSG,v}+\Delta {\dot{S}}_{COND}\right)$$

Finally, the irreversibilities generated in all the productive components of the combined cycle result by adding Equations (Equation 53), (Equation 56), and (Equation 57),
(58)$$\begin{array}{cc} {\dot{I}}_{prod}={\dot{I}}_{g}^{prod} & +{\dot{I}}_{v}^{prod}=\underset{\begin{array}{c} {\dot{I}}_{g}^{prod}:\;\mathrm{Contribution}\;\mathrm{to}\;\mathrm{the}\\ \;\mathrm{formation}\;\mathrm{process}\;\mathrm{of}\\ \mathrm{the}\;\mathrm{exhaust}\;\mathrm{gases}\; \end{array}}{\underset{︸}{{\dot{I}}_{GT}+{\dot{F}}_{HRSG}-\sum_{k\in HRSG}{\dot{E}}_{{\dot{Q}}_{k}}}}+\underset{\begin{array}{c} {\dot{I}}_{v}^{prod}:\;\mathrm{Contribution}\;\mathrm{to}\;\mathrm{the}\\ \mathrm{formation}\;\mathrm{process}\;\mathrm{of}\;\mathrm{the}\;\mathrm{waste}\\ \mathrm{heat}\;\mathrm{dissipated}\;\mathrm{in}\;\mathrm{the}\;\mathrm{condenser} \end{array}}{\underset{︸}{\sum_{k\in HRSG}{\dot{E}}_{{\dot{Q}}_{k}}-{\dot{P}}_{HRSG}+{\dot{I}}_{SC,prod}}}\\ = & T_{0}\left(\Delta {\dot{S}}_{tot,g}^{prod}-\frac{{\dot{m}}_{f}LHV}{T_{af}}+\sum_{k\in HRSG}\frac{{\dot{Q}}_{k}}{{\langle T\rangle }_{k}^{g}}\right)+T_{0}\left(\Delta {\dot{S}}_{tot,v}^{prod}-\sum_{k\in HRSG}\frac{{\dot{Q}}_{k}}{{\langle T\rangle }_{k}^{g}}\right) \end{array}$$

In agreement with the Gouy–Stodola theorem, the irreversibilities generated in the productive components along the formation process of the residues can be expressed in terms of the entropy generation in these components, i.e., ${\dot{I}}_{g}^{prod}=T_{0}{\dot{S}}_{gen,g}^{prod}$ and ${\dot{I}}_{v}^{prod}=T_{0}{\dot{S}}_{gen,v}^{prod}$. From Equation (Equation 58), it results that the entropy generation in the productive components that participates in the formation of the exhaust gases and the waste heat discarded in the condenser are respectively given by
(59a)$$\begin{array}{cc} {\dot{S}}_{gen,g}^{prod}=\Delta {\dot{S}}_{tot,g}^{prod}-\frac{{\dot{m}}_{f}LHV}{T_{af}}+\sum_{k\in HRSG} & \frac{{\dot{Q}}_{k}}{{\langle T\rangle }_{k}^{g}}=-\Delta {\dot{S}}_{stack,0}-\frac{{\dot{m}}_{f}LHV}{T_{af}}+\sum_{k\in HRSG}\frac{{\dot{Q}}_{k}}{{\langle T\rangle }_{k}^{g}} \end{array}$$
(59b)$$\begin{array}{cc} {\dot{S}}_{gen,v}^{prod}=\Delta {\dot{S}}_{tot,v}^{prod}-\sum_{k\in HRSG} & \frac{{\dot{Q}}_{k}}{{\langle T\rangle }_{k}^{g}}=-\Delta {\dot{S}}_{COND}-\sum_{k\in HRSG}\frac{{\dot{Q}}_{k}}{{\langle T\rangle }_{k}^{g}} \end{array}$$

As mentioned before, the combined cycle has two dissipative components: the stack and the condenser. The stack is a real component of which the objective is to expand as much as possible the plume of exhaust gases as well as to create a natural draft. For that purpose, additional exergy resources are required to operate auxiliary systems of induced and/or forced draft fans for getting rid of gases. However, since this work is only focussed on the estimation of the residue cost formation, the stack is assumed to be an imaginary dissipative component of which the resource and residue is ${\dot{E}}_{g13}$, and therefore, its irreversibility vanishes, ${\dot{I}}_{stack}=0$. The objective of the condenser is to condense water in order to remove the excess heat, to save water, and above all to perform a suction effect that causes the low-pressure turbine to work till very low pressures instead of till atmospheric pressure. The excess heat is, in fact, the residue to get rid of. Therefore, the fuel F of the condenser must be associated with the work of the pumping cooling water as well as the amount of evaporated water in the cooling tower.

The condenser is a heat exchanger in which the rejected heat generated by the phase change from wet steam to saturated liquid is transferred to the cooling water. In this sense, the resource of the condenser is the exergy change of the wet steam leaving the steam turbines and the saturated liquid water, and the product is exergy change of the cooling water. This productive purpose is achieved by the exchange of ${\dot{E}}_{{\dot{Q}}_{COND}}$—see Equation (Equation 17)—and the irreversibility of the overall condenser results then as
(60)$$\begin{array}{cc} {\dot{I}}_{COND,tot}= & {\dot{E}}_{v6}-{\dot{E}}_{v7}-{\dot{E}}_{{\dot{Q}}_{COND}}+{\dot{W}}_{CWP}=T_{0}\left(\Delta {\dot{S}}_{COND}+\frac{{\dot{Q}}_{COND}}{T_{COND}}\right)+{\dot{W}}_{CWP} \end{array}$$

The irreversibilities generated in the entire combined cycle are the sum of the irreversibilities of the productive and dissipative components. They are then obtained by adding to Equation (Equation 58) the irreversibilities of the stack and the condenser:(61)$$\begin{array}{cc} {\dot{I}}_{tot}={\dot{I}}_{tot}^{g}+{\dot{I}}_{tot}^{v}= & \underset{\begin{array}{c} {\dot{I}}_{tot}^{g}:\;\mathrm{Contribution}\;\mathrm{to}\;\mathrm{the}\;\mathrm{formation}\;\mathrm{process}\\ \mathrm{of}\;\mathrm{the}\;\mathrm{exhaust}\;\mathrm{gases}\; \end{array}}{\underset{︸}{{\dot{I}}_{GT}+{\dot{F}}_{HRSG}-\sum_{k\in HRSG}{\dot{E}}_{{\dot{Q}}_{k}}+{\dot{I}}_{stack}}}+\underset{\begin{array}{c} {\dot{I}}_{tot}^{v}:\;\mathrm{Contribution}\;\mathrm{to}\;\mathrm{the}\;\mathrm{formation}\;\mathrm{process}\\ \mathrm{of}\;\mathrm{the}\;\mathrm{waste}\;\mathrm{heat}\;\mathrm{dissipated}\;\mathrm{in}\;\mathrm{the}\;\mathrm{condenser} \end{array}}{\underset{︸}{\sum_{k\in HRSG}{\dot{E}}_{{\dot{Q}}_{k}}-{\dot{P}}_{HRSG}+{\dot{I}}_{SC,prod}+{\dot{I}}_{COND}}}\\ = & T_{0}\left(-\Delta {\dot{S}}_{stack,0}-\frac{{\dot{m}}_{f}LHV}{T_{af}}+\sum_{k\in HRSG}\frac{{\dot{Q}}_{k}}{{\langle T\rangle }_{k}^{g}}\right)+T_{0}\left(\frac{{\dot{Q}}_{COND}}{T_{COND}}-\sum_{k\in HRSG}\frac{{\dot{Q}}_{k}}{{\langle T\rangle }_{k}^{g}}\right) \end{array}$$

The irreversibilities of the productive components of the combined cycle and the entire combined cycle, Equations (Equation 58) and (Equation 61), can be both decomposed into two terms, corresponding to the irreversibilities that contribute to the formation of the exhaust gases and the waste heat dissipated in the condenser. These equations are examples of the application of the Gouy–Stodola theorem, and they clearly express that an entropy change is not always the same as entropy generation. This fact motivates to extend the criterion of entropy changes previously exposed by distributing the cost of the residues proportional to the irreversibilities generated through their formation process. In this criterion, the exhaust gases and waste heat cost allocation ratios for the productive components and those that referred only to the sum of the irreversibilities of the components involved in their formation are defined as follows
(62)$$\mu_{k}^{{\dot{I}}_{prod}}=\left\{\begin{array}{cc} \frac{{\dot{I}}_{k}}{{\dot{I}}_{g}^{prod}}, & k\in GT\\ \frac{{\dot{F}}_{HRSG,k}-{\dot{E}}_{{\dot{Q}}_{k}}}{{\dot{I}}_{g}^{prod}}, & k\in HRSG \end{array}\right.\;\;\mathrm{and}\;\;\beta_{k}^{{\dot{I}}_{prod}}=\left\{\begin{array}{cc} \frac{{\dot{E}}_{{\dot{Q}}_{k}}-{\dot{P}}_{HRSG,k}}{{\dot{I}}_{v}^{prod}}, & k\in HRSG\\ \frac{{\dot{I}}_{k}}{{\dot{I}}_{v}^{prod}}, & k\in SC_{prod} \end{array}\right.$$

Since the irreversibilities are always positive quantities, the cost allocation ratios are then also positive and between 0 and 1. These ratios satisfy that their summation over their associated components are all equal to one:(63)$$\sum_{k\in \Omega_{g}}\mu_{k}^{{\dot{I}}_{prod}}=\sum_{k\in \Omega_{v}}\beta_{k}^{{\dot{I}}_{prod}}=1$$

#### 3.3.3. Criterion of Distributed Exergy along the Process (C3)

Torres et al. formulated a criterion to estimate the residue cost allocation ratios based on the exergy distribution ratios in the system [1,2,31]. The main advantage of this criterion is that these ratios can be obtained directly from the information provided by the productive structure and the fuel-product table [28,32]. The FP table is a mathematical representation of a thermoeconomic model, representing the distribution of fuel and product through the power plant [28]. This table is expressed in terms of the resources and products exergy flows of each component and the cost distribution ratios. The columns and rows are associated with the resource and products of each component, and the cost distribution ratios appearing in the columns of the dissipative components correspond to residues cost allocation ratios.

This criterion requires identifying all residues, tracing the formation process of each residue, and locating its origin according to the productive structure of the plant [2]. The cost of the residue of the *j*th dissipative component should be then allocated to each productive component that feeds it in proportion to the amount of exergy it delivers to *j* [33] by means of the residue cost distribution ratios. These ratios are defined as the relations between the product exergy flow of the *i*th component used as resource by the *j*th dissipative component $\left({\dot{P}}_{i\to j}\right)$ and the sum of all the resources exergy flows fed with the product of the *i*th component $\left({\dot{F}}_{i\to k}\right)$,
(64)$$\psi_{ij}^{\langle \dot{E}\rangle }=\frac{{\dot{P}}_{i\to j}}{\sum_{k\in F_{i}}{\dot{F}}_{i\to k}}$$
where $F_{i}$ is the set of components that use the product of the *i*th component as resource on the basis of the productive structure of the plant. In other words, this is the set of components that distributes the product of the *i*th component over the components that are fed by this product.

Let us define $P_{j}$ as the set of all the productive components in which product exergy flows are used by the *j*th dissipative component as resources according to the productive structure. The residue cost distribution ratios defined by Equation (Equation 64) satisfy that, for a given dissipative component, the sum of all the components of $P_{j}$ is equal to one:(65)$$\sum_{i\in P_{j}}\psi_{ij}^{\langle \dot{E}\rangle }=1,\;\;\mathrm{for}\;\mathrm{the}\;j\mathrm{th}\;\mathrm{dissipative}\;\mathrm{component}$$

From the productive structure—see Figure 3—it can be observed that the exhaust gases dissipated in the stack have their origin in the exergetic stream g3, which at its time is produced by the combustion chamber and the compressor of the gas turbine, $P_{stack}=\left\{C,cc\right\}$. The products of these two productive components are both used as resources by the expansion turbine, the components of the HRSG, and the stack, i.e., $F_{C}=F_{cc}=\left\{t,stack\right\}\cup HRSG$. Based on Table 2, the following expression is obtained:(66)$$\sum_{k\in F_{i}}{\dot{F}}_{i\to k}={\dot{F}}_{t}+\sum_{k\in HRSG}{\dot{F}}_{k}+{\dot{F}}_{stack}={\dot{E}}_{g3},\;\;\mathrm{for}\;i\in P_{stack}=\left\{C,cc\right\}$$

The exhaust gase cost distribution ratios can then be determined from Equation (Equation 64), and as mentioned before, they correspond to the exhaust gases cost allocation ratios:(67)$$\mu_{i}^{\dot{E}}=\psi_{i,stack}^{\langle \dot{E}\rangle }=\frac{{\dot{P}}_{i\to stack}}{\sum_{k\in F_{i}}{\dot{F}}_{i\to k}}=\frac{{\dot{P}}_{i}}{{\dot{E}}_{g3}},\;\;\mathrm{for}\;i\in P_{stack}=\left\{C,cc\right\}$$

This criterion states that the exhaust gases are only formed by the exergy flows of the products of the compressor and the combustion chamber, $\mu_{C}^{\dot{E}}$, and $\mu_{cc}^{\dot{E}}$ respresents the contribution of these two productive components into their formation. By substituting the expressions of the products exergy flows into Equation (Equation 66), we may write
(68a)$$\begin{array}{cc} \mu_{C}^{\dot{E}}= & \frac{{\dot{E}}_{g2}-{\dot{E}}_{g1}}{{\dot{E}}_{g3}} \end{array}$$
(68b)$$\begin{array}{cc} \mu_{cc}^{\dot{E}}= & \frac{{\dot{E}}_{g3}-{\dot{E}}_{g2}}{{\dot{E}}_{g3}} \end{array}$$

Taking into account that ${\dot{E}}_{g1}=0$, it could be shown that $\mu_{c}^{\dot{E}}+\mu_{cc}^{\dot{E}}=1$.

The productive structure indicates that the product exergy flows of the components of the HRSG and the low-, intermediate-, and high-pressure pumps are used as resources by the condenser; in this way, $P_{COND}=HRSG\cup \left\{LPP,IPP,HPP\right\}$. Since the products of the components of $P_{COND}$ serve as resources to the low-, intermediate-, and high-pressure steam turbines as well as the condenser, then $F_{i}=\left\{LPST,IPST,HPST,COND\right\}$ for all $i\in P_{COND}$, and therefore from Table 2, it follows
(69)$$\begin{array}{c} \sum_{k\in F_{i}}{\dot{F}}_{i\to k}={\dot{F}}_{IPST}+{\dot{F}}_{HPST}+{\dot{F}}_{LPST}+{\dot{F}}_{COND}={\dot{E}}_{v1}-{\dot{E}}_{v2}+{\dot{E}}_{v4}-{\dot{E}}_{v5b}+{\dot{E}}_{v5}-{\dot{E}}_{v7},\\ i\in P_{COND}=HRSG\cup \left\{LPP,IPP,HPP\right\} \end{array}$$

The cost allocation ratios of the waste heat dissipated in the condenser are the same as the waste heat cost distribution ratios and can then be determined from Equation (Equation 64) according to
(70)$$\beta_{i}^{\dot{E}}=\psi_{i,COND}^{\langle \dot{E}\rangle }=\frac{{\dot{P}}_{i\to COND}}{\sum_{k\in F_{i}}{\dot{F}}_{i\to k}}=\frac{{\dot{P}}_{i}}{{\dot{E}}_{v1}-{\dot{E}}_{v2}+{\dot{E}}_{v4}-{\dot{E}}_{v5b}+{\dot{E}}_{v5}-{\dot{E}}_{v7}},\;i\in P_{COND}$$

The expressions of $P_{i}$, for $i\in P_{COND}$, can be obtained from Table 2. Using the exergetic balances in the LPD $\left({\dot{E}}_{v9}={\dot{E}}_{v9a}+{\dot{E}}_{v9b}+{\dot{E}}_{v9c}\right)$, the mixers M $\left({\dot{E}}_{v3}-{\dot{E}}_{v3a}={\dot{E}}_{v2}\right)$, and m $\left({\dot{E}}_{v5}={\dot{E}}_{v5a}+{\dot{E}}_{v5b}\right)$, it can be proved that $\sum_{i\in P_{COND}}\beta_{i}^{\dot{E}}=1$.

#### 3.3.4. Criterion of Distributed Entropy along the Process (C4)

Seyyedi and Farahat proposed a new criterion for the residue cost allocation based on the entropy distribution among the components of the system and not on the entropy generated along the process [16]. This criterion uses the criterion of distributed exergy through the processes, proposed by Torres et al. [2], as a starting point to internalize the residue cost generated by total exergy flows into the productive structure [34]. The criterion of distributed entropy along the process is based on the calculation of 1) the FP table; 2) the FP table using enthalpy (*H*) instead of exergy (*E*), FP${}^{\langle H\rangle }$ table; and 3) the FP table constructed by subtracting each element of the FP table from the corresponding element in the FP${}^{\langle H\rangle }$ table, FP${}^{\langle S\rangle }={\mathrm{FP}}^{\langle H\rangle }-\mathrm{FP}$, which represents the distribution of entropy through the power plant [16]. These three tables have associated three sets of distribution coefficients denoted as $\psi_{ij}^{\langle \dot{E}\rangle }$ (see Equation (Equation 64)), $\psi_{ij}^{\langle \dot{H}\rangle }$, and $\psi_{ij}^{\langle \dot{S}\rangle }$. The last two coefficients are given by
(71a)$$\begin{array}{cc} \psi_{ij}^{\langle \dot{H}\rangle } & =\frac{{\dot{P}}_{i\to j}^{\langle \dot{H}\rangle }}{\sum_{k\in F_{i}}{\dot{F}}_{i\to k}^{\langle \dot{H}\rangle }} \end{array}$$
(71b)$$\begin{array}{cc} \psi_{ij}^{\langle \dot{S}\rangle } & =\frac{\psi_{ij}^{\langle \dot{H}\rangle }{\dot{F}}_{j}^{\langle \dot{H}\rangle }-\psi_{ij}^{\langle \dot{E}\rangle }{\dot{F}}_{j}}{{\dot{F}}_{j}^{\langle \dot{S}\rangle }} \end{array}$$
where ${\dot{P}}_{i\to j}^{\langle \dot{H}\rangle }$ is the enthalpic term of ${\dot{P}}_{i\to j}$ and where ${\dot{F}}_{j}^{\langle \dot{H}\rangle }$ and ${\dot{F}}_{j}^{\langle \dot{S}\rangle }$ are respectively the enthalpic and entropic terms of ${\dot{F}}_{j}={\dot{F}}_{j}^{\langle \dot{H}\rangle }-{\dot{F}}_{j}^{\langle \dot{S}\rangle }$. Proceeding analogously as in Section 3.3.3, the residues distribution ratios associated with the FP${}^{\langle \dot{H}\rangle }$ table and determined from Equation ([Disp-formula FD71a-entropy-22-00299]) are expressed as follows:
(72a)$$\begin{array}{cc} \; & \;\;\;\;\;\;\;\;\;\;\;\;\;\;\;\;\;\;\;\;\;\;\;\;\;\;\;\;\;\;\psi_{i,stack}^{\langle \dot{H}\rangle }=\frac{{\dot{P}}_{i}^{\langle \dot{H}\rangle }}{{\dot{H}}_{g3}-{\dot{H}}_{0}},\;i\in P_{stack}=\left\{C,cc\right\} \end{array}$$
(72b)$$\begin{array}{cc} \psi_{i,COND}^{\langle \dot{H}\rangle } & =\frac{{\dot{P}}_{i}^{\langle \dot{H}\rangle }}{{\dot{H}}_{v1}-{\dot{H}}_{v2}+{\dot{H}}_{v4}-{\dot{H}}_{v5b}+{\dot{H}}_{v5}-{\dot{H}}_{v7}},\;i\in P_{COND}=HRSG\cup \left\{LPP,IPP,HPP\right\} \end{array}$$

The residue cost allocation ratios for the criterion of distributed entropy along the process are derived by combining Equations (71b), (67), and (70) to obtain
(73a)$$\begin{array}{cc} \; & \;\;\;\;\;\;\;\;\;\;\;\;\;\;\;\;\mu_{i}^{\langle \dot{S}\rangle }=\psi_{i,stack}^{\langle \dot{S}\rangle }=\frac{\psi_{i,stack}^{\langle \dot{H}\rangle }{\dot{F}}_{stack\to 0}^{\langle \dot{H}\rangle }-\mu_{i}^{\dot{E}}{\dot{F}}_{stack}}{{\dot{F}}_{stack\to 0}^{\langle \dot{S}\rangle }},\;i\in P_{stack}=\left\{C,cc\right\} \end{array}$$
(73b)$$\begin{array}{cc} \beta_{i}^{\langle \dot{S}\rangle }= & \psi_{i,COND}^{\langle \dot{S}\rangle }=\frac{\psi_{i,COND}^{\langle \dot{H}\rangle }{\dot{F}}_{COND}^{\langle \dot{H}\rangle }-\beta_{i}^{\dot{E}}{\dot{F}}_{COND}}{{\dot{F}}_{COND}^{\langle \dot{S}\rangle }},\;i\in P_{COND}=HRSG\cup \left\{LPP,IPP,HPP\right\} \end{array}$$
where ${\dot{F}}_{stack\to 0}^{\langle \dot{S}\rangle }=T_{0}\Delta {\dot{S}}_{stack,0}=T_{0}\left({\dot{S}}_{g13}-{\dot{S}}_{cg0}\right)$ and ${\dot{F}}_{COND}^{\langle \dot{S}\rangle }=T_{0}\Delta {\dot{S}}_{COND}=T_{0}\left({\dot{S}}_{v7}-{\dot{S}}_{v6}\right)$.

## 4. Results

### 4.1. Thermodynamics of the Combined Cycle

The thermodynamic states at the actual operating condition of the triple-level pressure combined cycle with a 2×2×1 arrangement are presented in Table 5. They are obtained by solving the energy and mass balance system of equations, given by Equation (Equation 1)–(Equation 27), by using the information of Table 1 and steam tables [35]. To determine the specific exergy, the adopted dead state corresponds to $T_{0}=25$ °C and $P_{0}=1.013$ bar.

The thermodynamic performance indicators for the gas turbine, the steam cycle, and the overall combined cycle are computed from the thermodynamic states of Table 5 together with the equations of Section 2, and they are summarized in Table 6.

One gas turbine provides 139 MW of net power output using 313 kg${}_{\mathrm{air}}$/s and burning 7.68 31 kg${}_{f}$/s of natural gas. In other words, the GT requires 0.1986 kg${}_{f}$ of natural gas to produce 1 kWh of net power with a thermal efficiency of 35.16%. The GT produces 321.08 kg${}_{\mathrm{cg}}$/s of combustion gases at $T_{g4}=617.58$ °C with an enthalpy flow rate of ${\dot{H}}_{g4}-{\dot{H}}_{cg0}=481.93$ MW.

In the HRSG, 387 MW of thermal energy is transferred from the combustion gases to the water system to generate steam at high (v1), intermediate (v4), and low (v5a) pressures. The steam generation uses only 79.1 % of the thermal energy of the combustion gases leaving the gas turbine; the other 20.9% is delivered to the environment through the exhaust gases dissipation.

The HRSG has a approach temperature difference of $T_{g4}-T_{v1}=91.77C$. This value is within the range of hot approach temperature differences reported by Ganapathy, which corresponds to 55.56 °C–195 °C [36]. In this subsytem, the greatest heat transfer takes places in the high-pressure superheater and intermediate pressure reheater (HPSH+IPRH), the high-pressure evaporator (HPEV), and the high-pressure economizer (HPEC) corresponding to 12.9%, 11.6%, and 7.0%, respectively. The low-pressure superheater (LPSH) and the intermediate pressure superheater (IPSH) are the components with the lowest heat transfer, 0.7%, and 0.2%, respectively. These last two components have the lowest temperature differences between the gas turbine exhaust gases and the generated steam, 52.78 °C and 13.76 °C, respectively. This fact indicates the pinch-point temperature differences are in the correct operating range and avoid the existence of an intersection point between the temperature profiles of the combustion gases and water.

The mass flow rate of the high-pressure steam generated in the HRSG is ${\dot{m}}_{HP}=76.77\;{\mathrm{kg}}_{s}$/s, and it is fed to the high-pressure steam turbine (HPST), of which the pressure ratio is 3.97, to produce 26.97 MW. In the mixer M, the steam leaving the HPST (v2) is mixed with the steam exiting the IPSH (v3a) to obtain intermediate-pressure steam (v3) with a mass flow rate of ${\dot{m}}_{IP}=11.28\;{\mathrm{kg}}_{s}$/s. This steam is used by the IPST, with a pressure ratio of 9.08, to produce ${\dot{W}}_{HPST}=48.75$ MW. Finally, the steam leaving the IPST (v5b) is mixed with the generated steam in the LPSH (v5a) to get the low-pressure steam (v5), which is supplied to the LPST. The LPST has a pressure ratio of 45.25 and provides 61.80 MW. The low-, intermediate-, and high-pressure pumps are electrically driven with electricity coming from the steam cycle electric generator. This means that one part of the power generated by the steam cycle is used by the pumps, corresponding to 1.29 MW. The net power output of the steam cycle is then 134.48 MW; this means that 2.05 ${\mathrm{kg}}_{s}$ of steam are required to produce 1 kWh of power. At the actual operating condition, the output power generated by the combined cycle with a 2 × 2 × 1 arrangement is 412.88 MW burning 15.36 ${\mathrm{kg}}_{f}$/s of natural gas, and its computed thermal efficiency is 54.30%.

### 4.2. Exergy Analysis

The exergy content of the combined cycle resource is ${\dot{E}}_{f}=664.79$ MW (100%). From this exergetic resource, 62.10% is used to generate the net output power of the combined cycle $\left({\dot{W}}_{mCC}=412.88\;\mathrm{MW}\right)$, 3.49% goes to the exergy flow of the residual streams $\left({\dot{R}}_{CC}=23.21\;\mathrm{MW}\right)$, 34.13% corresponds to the irreversibility of the combined cycle $\left({\dot{I}}_{CC}=226.94\;\mathrm{MW}\right)$, and 0.26% serves to generate the power for driving the cooling water pump in the condenser (${\dot{W}}_{CWP}=1.75$ MW). The gas turbine contributes to generate 67.42% of the combined cycle net output power, and the remaining part corresponds to the contribution of the steam cycle. The exergy of the exhaust gases released in the stack is ${\dot{E}}_{g13}=10.71$ MW and represents 46.15% of the combined cycle residual exergy flow, while the rest is associated to the exergy of the heat flow dissipated from the condenser, ${\dot{E}}_{{\dot{Q}}_{COND}}=12.50$ MW.

The irreversibilities of the combined cycle can be understood as the accumulation of the irreversibilities generated in all the combined cycle components. The GT, HRSG, and SC contribute respectively with 74.59%, 19.25%, and 6.16% to ${\dot{I}}_{CC}$. The exergetic efficiencies of these subsystems are respectively 74.54%, 74.95%, and 82.65%. In a combined cycle, the exergetic efficiency of the gas turbine increases because its product is not only the output power but also the exergy flow of the combustion gases (g4). Table 7 shows the highest irreversibilities are generated in CC, C, and t, corresponding respectively to 56.32%, 11.29%, and 6.98% of ${\dot{I}}_{CC}$. For the HRSG, the IPRH+HPSH is the section that contributes the most to ${\dot{I}}_{HRSG}$, with 8.9 MW while the IPSH is the section that has the least contribution to the HRSG irreversibilities, with 0.16 MW. The LPST is the component of the SC with the highest irreversibility, 8.0 MW. The irreversibilities of the low-, intermediate-, and high-pressure drums are all zero because they are expressed only in terms of the water physical exergy which does not change through these components; see Table 5. Since they do not present irreversibilities, then their exergetic efficiency is 100% and their exergoeconomic operation costs are also zero.

The irreversibilities of the combined cycle can also be conceived as the sum of the irreversibilities generated through the formation process of the exhaust gases and the heat dissipated from the condenser, which are formed to generated the power in the gas an steam cycles. In other words, these irreversibilities are generated by the processing of air and combustion gases in the GT and HRSG (${\dot{I}}_{g}$) as well as steam in the HRSG and steam cycle (${\dot{I}}_{v}$). The irreversibility generated from the gas side (${\dot{I}}_{g}={\dot{I}}_{GT}+{\dot{I}}_{HRSG,g}+{\dot{I}}_{stack}=184.60$ MW), corresponding to 81.34% of ${\dot{I}}_{CC}$, accompanies the formation of the exhaust gases, and that from the steam side (${\dot{I}}_{v}={\dot{I}}_{HRSG,v}+{\dot{I}}_{SC}=42.34$ MW), representing 18.66% of ${\dot{I}}_{CC}$, runs parallel to the formation of the heat dissipated from the condenser. In the HRSG, the resource exergies of the side gas are greater than those of the steam side while the irreversibilities of the gas side are lower than those of steam side, except for the IPSH, IPEV, and IPEC in both cases. For these six components of the HRSG, this fact derives on higher exergetic efficiencies for the gas side in comparison with the steam side, as shown in Table 7. For the other three components, the exergetic efficiencies present an opposite behavior.

### 4.3. Exergetic Costs Analysis

#### 4.3.1. Exergetic Cost of Streams

The criterion C1 is based on the entropy changes of productive components associated with the residues formation. The temperature–entropy diagrams of Figure 4 for the open gas turbine (gas side) and closed steam cycle (steam side) indicate that the entropy changes of the productive components go together with the exhaust gases and the waste heat formation respectively.

For the exhaust gases, as shown in Table 8, the residue cost allocation ratios for the compressor, the combustion chamber, and the expansion turbine are positive values since their entropy changes are positive while the components of the HRSG are all negative values because their entropy changes are negative. The combustion chamber is the productive component with the highest entropy change, and therefore, it is the component that more contributes to the cost of the exhaust gases, $\mu_{cc}^{C1}=1.992$.

For the heat dissipated from the condenser, the residue cost allocation ratios for the compressor, the combustion chamber, and the expansion turbine are equal to zero, since these components are not involved in this residue formation. The waste heat allocation ratios are positive for all the productive components participating in its formation process. Under this criterion, the HRSG contributes to the waste heat formation in 93.98% and the rest is attributed to the productive components of the SC. Among the HRSG components, the HPEV, HPSH+IPRH, LPEC, LPEV, and HPEC, listed in increasing order of β (see Table 8), are the components that mostly contribute to this residue formation, with 81.00%. This behavior is related to the fact that the high-pressure sections of the HRSG handle the highest pressures and temperatures of steam and, therefore, the lowest specific entropy changes (see Table 5 and Figure 4b); despite this, the mass flow rates of the HPEV and HPSH+IPRH are relatively high (${\dot{m}}_{HPEV}={\dot{m}}_{HPSH}=76.77$ kg${}_{s}$/s and ${\dot{m}}_{IPRH}=80.06$ kg${}_{s}$/s). In the LPEV, 21.99 kg${}_{s}$/s of compressed liquid water is brought to saturated steam at T=139.16 °C and P=3.53 bar, with an evaporation specific entropy change of 5.21 kJ/(kg K). The total entropy change in the LPEV is such that it is in the third place in the formation of the waste heat (see Table 7).The LPST, IPST, and HPST are the productive components of the SC of which participation in the waste heat formation (3.44%, 1.70%, and 0.80%, respectively, see Table 8) is the most important. In these three components, the changes of specific entropy are very low; however, its order of contribution is related to the steam flows handled by the turbines (the greatest steam flow is expanded in the LPST).

For this criterion, in one hand, Table 9 shows that the exergetic costs of the gas streams (g2 to g13) are greater than those of criteria C2, C3, and C4. On the other hand, for the steam streams, the costs are lower than the other three criteria, except for the stream v15, for which it is fulfilled that $E_{v15}^{*C3}<E_{v15}^{*C1}<E_{v15}^{*C2}<E_{v15}^{*C4}$.

According to the cost balances for the productive components, given by Equation (Equation 40), the residue cost is charged to a productive component as long as the residue cost allocation ratios are positive, as is for the case of the gas turbine components. Otherwise, for negative residue cost allocation ratios, the residue costs represent a saving for the component, as for example for the HRSG components on the gas side ($\mu_{HRSG,K}^{C1}<0$). In this study case, the exergetic costs of the products of the GT components computed from the criterion C1 are higher than those obtained from the other criteria. The product costs of the HRSG and steam cycle components, derived from this criterion, are the lowest with respect to the criteria C2, C3, and C4 except for the IPSH, for which it is satisfied that $P_{IPSH}^{*C2}<P_{IPSH}^{*\mathrm{CI}}<P_{IPSH}^{*C4}<P_{IPSH}^{*C3}$.

In general, it can be observed from Table 9 that the liquid water and steam streams have the highest exergy costs for the criteria C3 and C4, while the lowest costs are always associated to the criterion C1.

For the criterion C2, the cost of residues is in direct proportion to the irreversibility or exergy destruction of the productive components. Since the irreversibility is a nonnegative thermodynamic property, the residue cost allocated to the productive components are then greater than or equal to zero. This means that the cost of a residue allocated to a productive component is always charged to its product cost. As for the criterion C1, the components of the GT and the HRSG (gas side) are the productive components which are responsible for the exhaust gases formation. The combustion chamber is the productive component with the highest irreversibility, and therefore, it is the component that more contributes, with 69.27%, to the formation cost of the exhaust gases ($E_{g13}^{*}$). The compressor participates with 13.88%, while the components of the HRSG on the gas side contribute with 8.24% to the formation cost of this residue. The HRSG components of the water–steam circuit and the productive components of the SC contribute respectively 67.06% and 32.93% to the formation of the waste heat dissipated from the condenser. According to Table 8, the waste heat is mainly allocated to the LPST, IPRH+HPSH, and HPEV in proportions of 18.84%, 17.22%, and 14.35% respectively. Table 9 indicates the exergetic costs of the gas streams (g2 to g13) are higher than those obtained by the criteria C1, C3, and C4. For the streams of water and steam, the cost is greater than those computed by using the criterion C1 ($E^{*C1}<E^{*C2}$).

In the criterion of distributed exergy (C3), the cost of residues is allocated only to those productive components from which the resource of the dissipative component has originated. In this study case, the resource of the stack (${\dot{E}}_{g13}$) corresponds to the combustion gases, produced by the compressor and the combustion chamber; and their cost allocation ratios are respectively $\mu_{C}^{C3}=0.32$ and $\mu_{cc}^{C3}=0.68$. The components of the HRSG and the pumps (HPP, IPP, and LPP) are the productive components involved in waste heat formation. The HRSG contributes 99.30% to the cost of this residue, while the allocation of the pumps to the cost of this residue is negligible. The product exergy flows of the HPSH+IPRH, HPEV, and HPEC represent 75.53% of the resource exergy flow provided to the condenser.

The criterion C4 is based on the entropy distribution in the productive components that feed a dissipative component. In this criterion, the difference between the enthalpy and exergy distribution of the components of which the products serve as a resource to the dissipative one, as established by Equation (71b), is related to the entropy change in the dissipative component. Since this criterion is based on the enthalpy and exergy fuel-product tables, the exhaust gases cost allocation ratios are only defined for the compressor and combustion chamber, $\mu_{C}^{C4}=0.262$ and $\mu_{cc}^{C4}=0.738$. As for the criterion C3, the HRSG components and the pumps are the components that participate in the formation of the waste heat dissipated from the condenser. As it can be observed from Table 8, in this criterion, the HRSG is the subsystem that mostly impacts the formation cost of the waste heat dissipated from the condenser, with 99.36%, followed by criteria C1 (93.98%), C2 (67.06%), and C3 (98.94%).

In this criterion, Table 9 shows that the exergetic costs of streams g3 to g13 are equal to the costs computed from criterion C3, except for the compressed air stream (g2), for which it is fulfilled that $E_{g2}^{*C4}<E_{g2}^{*C3}$. The cost obtained from the criteria C1 and C2 enclose those of criteria C3 and C4, as follows:
(74a)$$\begin{array}{cc} E_{i}^{*C2}<E_{i}^{*C4}=E_{i}^{*C3} & <E_{i}^{*C1},\;\;i=g3,\ldots ,g13 \end{array}$$
(74b)$$\begin{array}{cc} E_{g2}^{*C2}<E_{g2}^{*C4} & <E_{g2}^{*C3}<E_{g2}^{*C1} \end{array}$$

#### 4.3.2. Product Exergetic Costs of the Productive Components

For all the evaluated criteria, the components of the gas turbine present the highest exergetic costs for the resources and products, as shown in Table 10. The gas turbine cycle provides one of the two products of the combined cycle, the net output power of the GT. Among the four criteria, the criteria C1 and C2 conduce respectively to the highest and lowest product costs for the components of the gas turbine:(75)$$E_{{\dot{W}}_{mGT}}^{*C2}<E_{{\dot{W}}_{mGT}}^{*C4}=E_{{\dot{W}}_{mGT}}^{*C3}<E_{{\dot{W}}_{mGT}}^{*C1}$$

The products cost of the HRSG components are the lowest for the criterion C1, since the products of these components are conformed by liquid water and steam streams, which present the lowest exergetic costs. These product costs are the highest for the criteria C3 or C4, and for the criterion C2, these costs are comprised of that between those of criteria C1 and C3 or C4.

As for the HRSG, the product costs of the SC components computed by using the criterion C1 are the lowest ones. Nevertheless, there is not a general criterion maximizing these costs. For some components, such as LPST, LPP, and EG, the criterion C2 is the one that yields to the greatest product costs while, for the HPST, IPST, IPP, HPP, and M, the criterion C3 provides the highest costs of products.

The net output power of the SC is the other product of the combined cycle and in accordance with Table 9, the exergetic cost of this product computed from the criteria C1, C2, C3, and C4 maintain the following relation:(76)$$E_{{\dot{W}}_{mSC}}^{*C1}<E_{{\dot{W}}_{mSC}}^{*C4}=E_{{\dot{W}}_{mSC}}^{*C3}<E_{{\dot{W}}_{mSC}}^{*C2}$$
This equation indicates that the criterion proposed in this work yields to an upper limit for the cost of the SC net output power.

#### 4.3.3. Residue Formation Cost

For the criteria C1 and C2, the formation cost of the residues is allocated to all the productive components; however, for the criteria C3 and C4, this cost is only allocate to the productive components that originate the residues.

In the criteria C3 and C4, the formation cost of the exhaust gases is only imputed to the combustion chamber and the compressor. As shown in Figure 2a and Figure 3, it is in these two components that the exergic flow ${\dot{E}}_{g3}$ originated. A part of this exegy stream serves as a resource to the expansion turbine for producing the net output power of the gas turbine and to the HRSG components for producing heat to generate steam, while the remaining part is not useful exergy and gives place to the residual exergy ${\dot{E}}_{g13}$. In these criterion, the formation cost of the waste heat dissipated from the condenser is only accounted for in the components of the HRSG and the pumps of the steam cycle because they are the productive components that originate useful exergy to produce the steam turbines power and unuseful exergy consumed in the condenser to generate the residue of the steam cycle.

Two types of productive components can be then distinguished: the provider and consumer productive components. The provider productive components originate an exergy stream, and a part if this exergy flow is used as resource by the consumer productive components to fulfill the productive purpose of the energy system. The more exergy the comsumer productive components use, the lower the exergy content of the residual streams. The criteria C3 and C4 allocate then the formation cost of the exhausted gases to the provider productive components.

Table 10 and Figure 7 show that, for all the criteria, the combustion chamber has the greatest responsibility on the formation cost of the exhaust gases. The compressor has also a significant responsibility on the formation cost of this residue; however, it has greater responsibility for criteria C3 and C4 than the other two approaches. For criteria C1 and C2, the expansion turbine occupies the third place, in order of importance, in the contribution to the formation cost of this residue. The exhaust gase formation costs allocated to components of the HRSG are all negative for the criterion C1, favoring the reduction of the product cost of each of these components and compensating the excessive influence of the combustion chamber on the residue formation cost. In criterion C2, the formation cost of the exhaust gases allocated to the HRSG components are all positive, indicating that these components are also responsible for the formation cost of this residue. The LPEC and IPSH are components of the HRSG with the greatest and least responsibilities in the formation cost of this waste. Despite the mentioned differences, the exergetic formation cost of the exhaust gases is between 14.41 MW and 14.85 MW; the lower and upper limits correspond to the criteria C2 and C1 respectively. For the different evaluated criteria, the exhaust gase formation cost obeys then the following order relation
(77)$$E_{g13}^{*C2}<E_{g13}^{*C4}=E_{g13}^{*C3}<E_{g13}^{*C1}$$

For the four evaluated criteria, the HPSH+IPRH and HPEV are the components of the HRSG with the greatest responsibility on the formation cost of the waste heat dissipated in the condenser, as it can be observed in Table 10 and Figure 7. In the same way, the low- and intermediate-pressure pumps are the components of the SC with the smallest responsibility in the formation cost of this residue. Nevertheless, for the criteria C2, the low-pressure steam turbine is the component with the major contribution on the formation cost of the waste heat, since in this criteria, the hierarchy is proportional to irreversibilities generated in each productive component. For the criteria evaluated in this work, the exergetic formation cost of the waste heat dissipated from the condenser goes from 26.85 MW for the criterion C1 to 28.22 MW for the criterion C4, and the order relation between all the criteria is given by
(78)$$E_{{\dot{Q}}_{COND}}^{*C1}<E_{{\dot{Q}}_{COND}}^{*C2}<E_{{\dot{Q}}_{COND}}^{*C3}<E_{{\dot{Q}}_{COND}}^{*C4}$$

## 5. Conclusions

The three-pressure combined cycle has two residues: the physical exergy of the exhaust gases and the exergy associated to the heat dissipated from the condenser into the environment. An alternative criterion has been proposed to allocate the residue formation cost to the productive components involved in its formation. This criterion is based on the irreversibilities generated in each productive component participating in the formation of a residue. Under this approach, the physical exergy of exhaust gases is formed in all the productive components of the gas turbine and the heat recovery steam generator. The heat dissipated from the condenser is formed in all productive components of the heat recovery steam generator and those of the steam cycle. The proposed criterion is an extension of the entropy change criterion because, in addition to the entropy changes, it also includes the exergy flows associated to the heat transfer through the boundaries of the energy system. In these two criteria, all the productive components of each circuit intervene in the formation of their respective residue, unlike the other two criteria based on distribution coefficients, in which the productive components participating in the residues formation are only those that originate them.

The main disadventage of the way that the production cost is computed in this work is that it does not permit to calculate the costs of internal streams for assessing of the impact of the additional fuel required to compensate component malfunctions.

## Figures and Tables

**Figure 1 entropy-22-00299-f001:**
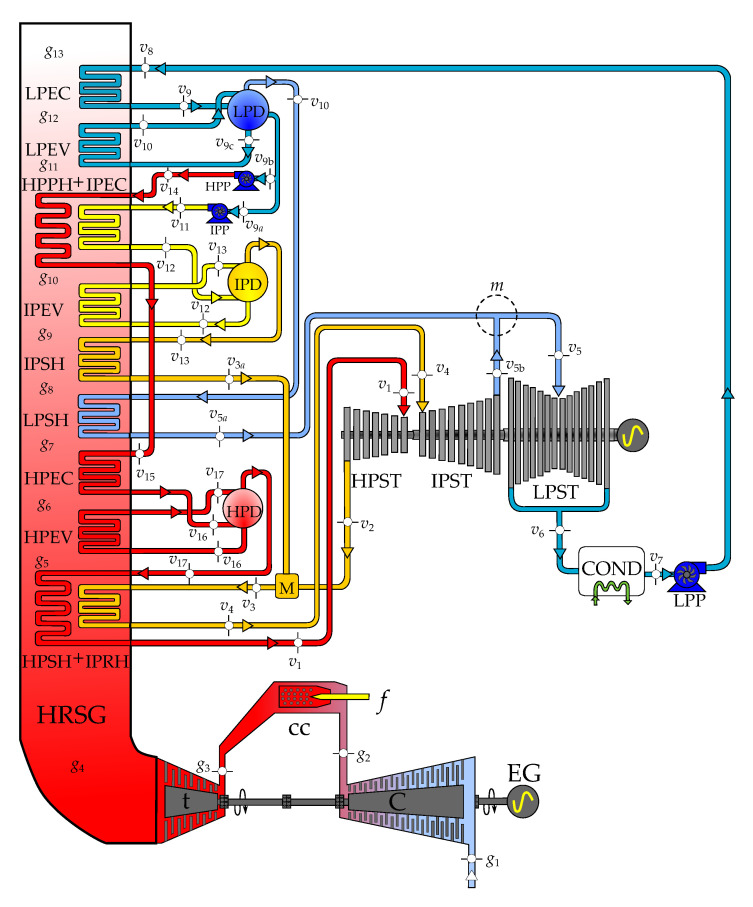
Schematic diagram of a triple-pressure-level combined cycle.

**Figure 2 entropy-22-00299-f002:**
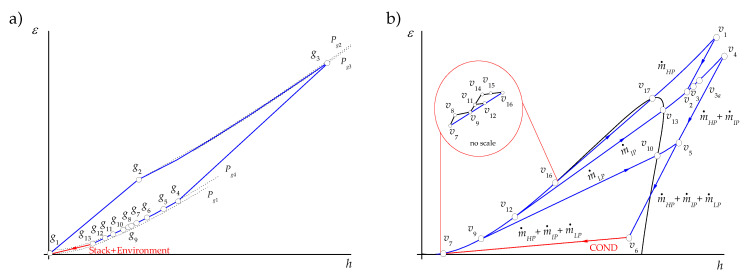
Exergy–enthalpy diagram of the (**a**) gas turbine cycle and (**b**) steam cycle.

**Figure 3 entropy-22-00299-f003:**
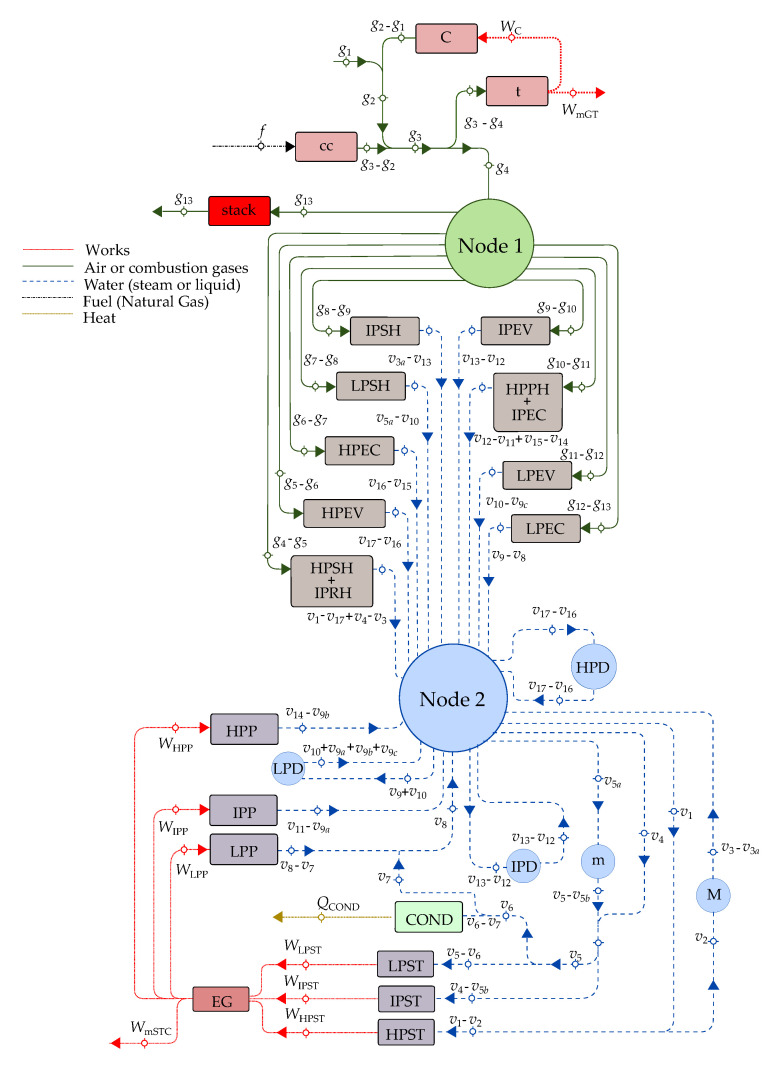
Productive structure of the combined cycle power plant.

**Figure 4 entropy-22-00299-f004:**
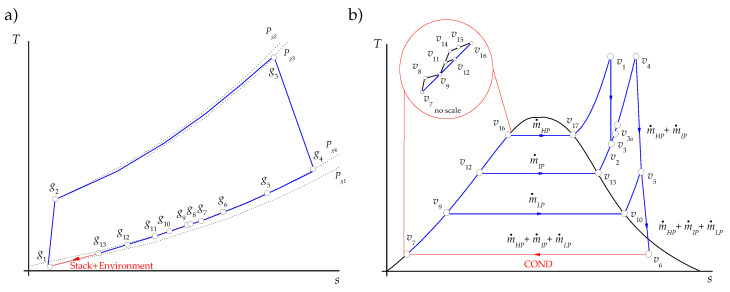
Temperature–entropy diagram of the (**a**) gas turbine cycle and (**b**) steam cycle.

**Figure 5 entropy-22-00299-f005:**
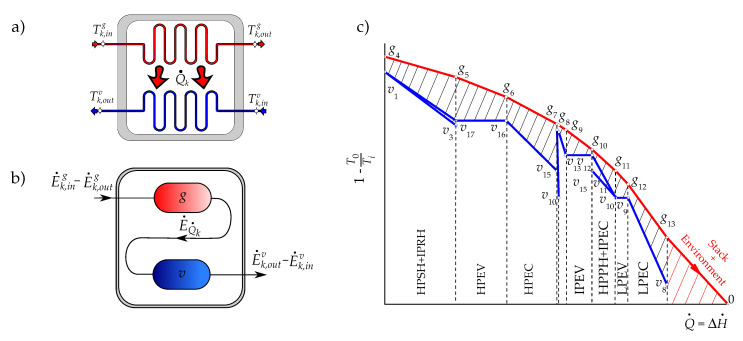
(**a**) The *k*th component of the HRSG, (**b**) productive structure of a component of the HRSG, and (**c**) the Carnot coefficient profile as a function of the recovered heat in the HRSG.

**Figure 6 entropy-22-00299-f006:**
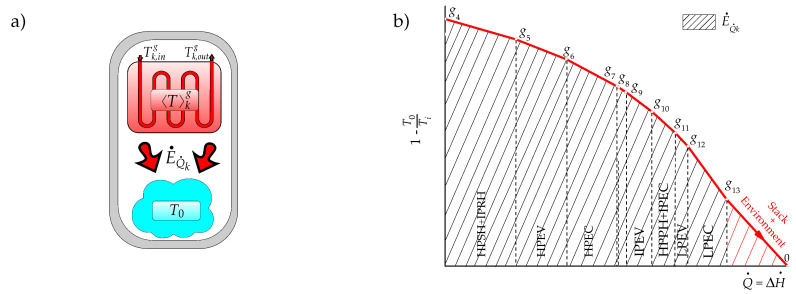
(**a**) Schematic diagram of the heat exergy flow of a thermal energy reservoir with nonconstant temperature and (**b**) geometric interpretation of the heat exergy flow of a thermal energy reservoir with nonconstant temperature in the Carnot coefficient-recovered heat in the HRSG diagram.

**Figure 7 entropy-22-00299-f007:**
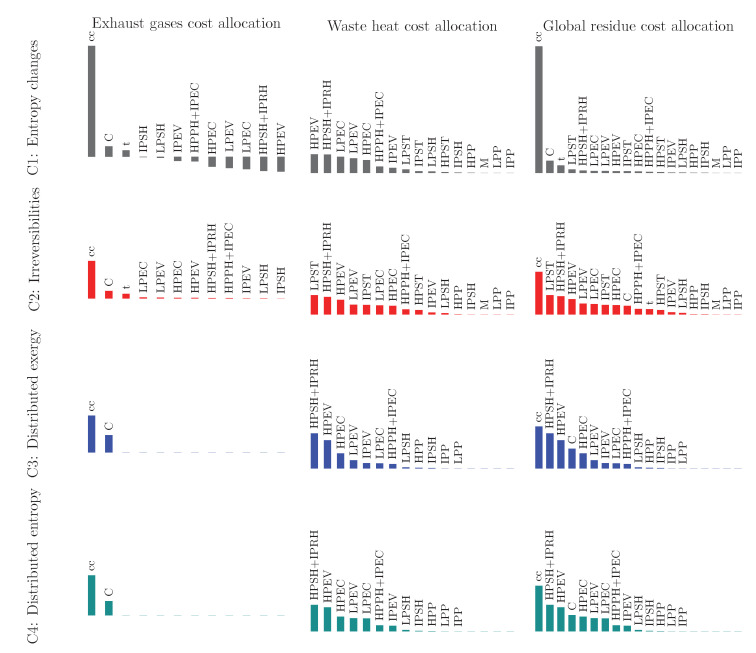
Contribution of the combined cycle components, in order of importance, in the exergetic formation cost of the exhaust gases, waste heat dissipated from the condenser, and global residue for the residue cost allocation criteria of entropy changes (C1: ■), irreversibilities (C2: ■), distributed exergy (C3: ■), and distributed entropy (C4: ■).

**Table 1 entropy-22-00299-t001:** Characteristics of the gas turbine and steam cycle, fuel, ambient conditions, pinch-point temperature differences in the heat recovery steam generator, and combustion parameters.

**Gas Turbine**								**Fuel**					**Ambient Conditions**			
${\dot{W}}_{mGT}$	TIT	$\pi_{C}$	$\eta_{C}$	$\eta_{t}$	$\eta_{EG}$	$\Delta P_{cc}$	$\Delta P_{t}$		$X_{{\mathrm{CH}}_{4}}$	$X_{C_{2}H_{6}}$	$X_{C_{3}H_{8}}$		$T_{a}$	$P_{a}$	ϕ
(MW)	(°C)	(-)	(-)	(-)	(-)	(%)	(%)		(%)	(%)	(%)		(°C)	(bar)	(%)
139.2	1300	16	0.88	0.9	1.0	2	1		88	9	3		25	1.013	45
													**Pinch-point temperature**			
**Steam cycle**												**differences, $\Delta T_{pp}$**			
$T_{v1}$	HP	IP	LP	$P_{COND}$	$\eta_{ST}$	$\eta_{P}$	$\eta_{CWP}$	$\Delta T_{COND}$		$T_{CW1}$	$T_{CW2}$		LPEV	IPEV	HPEV
(°C)	(bar)	(bar)	(bar)	(bar)	(-)	(-)	(-)	(°C)		(°C)	(°C)		(°C)	(°C)	(°C)
525.8	106.3	25.2	4.1	0.08	0.88	0.85	0.60	15		30.23	30.18		22.37	15	84.51
**Combustion parameters of Equation (Equation 9)**															
*m*	*n*	$X_{stHA}$	$X_{DA}$	$X_{H_{2}O}$	λ	$\alpha_{1}$		$\alpha_{2}$		$\alpha_{3}$	$\alpha_{4}$		$\alpha_{5}$	$\alpha_{6}$	$\alpha_{7}$
4.3	1.15	10.74	0.986	0.014	1.396	1.139		2.510		20.047	3.106		0.010	4.77 × $10^{-4}$	0.011

**Table 2 entropy-22-00299-t002:** Exergy flow rates of the resource, product, and irreversibilities of the productive components of the combined cycle.

Component	$\dot{F}$	$\dot{P}$	$\dot{I}$
**Gas turbine cycle**				
C	${\dot{W}}_{C}$	${\dot{E}}_{g2}-{\dot{E}}_{g1}$	$T_{0}\left({\dot{S}}_{g2}-{\dot{S}}_{g1}\right)$
cc	${\dot{E}}_{f}$	${\dot{E}}_{g3}-{\dot{E}}_{g2}$	$T_{0}\left({\dot{S}}_{g3}-{\dot{S}}_{g2}\right)-{\dot{m}}_{f}LHV\frac{T_{0}}{T_{af}}$
t	${\dot{E}}_{g3}-{\dot{E}}_{g4}$	${\dot{W}}_{C}+{\dot{W}}_{mGT}$	$T_{0}\left({\dot{S}}_{g4}-{\dot{S}}_{g3}\right)$
**Heat recovery steam generator**				
HPSH+IPRH	${\dot{E}}_{g4}-{\dot{E}}_{g5}$	${\dot{E}}_{v1}-{\dot{E}}_{v17}+{\dot{E}}_{v4}-{\dot{E}}_{v3}$	$T_{0}\left({\dot{S}}_{g5}-{\dot{S}}_{g4}\right)$
			$+T_{0}\left({\dot{S}}_{v1}-{\dot{S}}_{v17}+{\dot{S}}_{v4}-{\dot{S}}_{v3}\right)$
HPEV	${\dot{E}}_{g5}-{\dot{E}}_{g6}$	${\dot{E}}_{v17}-{\dot{E}}_{v16}$	$T_{0}\left({\dot{S}}_{g6}-{\dot{S}}_{g5}+{\dot{S}}_{v17}-{\dot{S}}_{v16}\right)$
HPEC	${\dot{E}}_{g6}-{\dot{E}}_{g7}$	${\dot{E}}_{v16}-{\dot{E}}_{v15}$	$T_{0}\left({\dot{S}}_{g7}-{\dot{S}}_{g6}+{\dot{S}}_{v16}-{\dot{S}}_{v15}\right)$
LPSH	${\dot{E}}_{g7}-{\dot{E}}_{g8}$	${\dot{E}}_{v5a}-{\dot{E}}_{v10}$	$T_{0}\left({\dot{S}}_{g8}-{\dot{S}}_{g7}+{\dot{S}}_{v5a}-{\dot{S}}_{v10}\right)$
IPSH	${\dot{E}}_{g8}-{\dot{E}}_{g9}$	${\dot{E}}_{v3a}-{\dot{E}}_{v13}$	$T_{0}\left({\dot{S}}_{g9}-{\dot{S}}_{g8}+{\dot{S}}_{v3a}-{\dot{S}}_{v13}\right)$
IPEV	${\dot{E}}_{g9}-{\dot{E}}_{g10}$	${\dot{E}}_{v13}-{\dot{E}}_{v12}$	$T_{0}\left({\dot{S}}_{g10}-{\dot{S}}_{g9}+{\dot{S}}_{v13}-{\dot{S}}_{v12}\right)$
HPPH+IPEC	${\dot{E}}_{g10}-{\dot{E}}_{g11}$	${\dot{E}}_{v15}-{\dot{E}}_{v14}+{\dot{E}}_{v12}-{\dot{E}}_{v11}$	$T_{0}\left({\dot{S}}_{g11}-{\dot{S}}_{g10}\right)$
			$+T_{0}\left({\dot{S}}_{v15}-{\dot{S}}_{v14}+{\dot{S}}_{v12}-{\dot{S}}_{v11}\right)$
LPEV	${\dot{E}}_{g11}-{\dot{E}}_{g12}$	${\dot{E}}_{v10}-{\dot{E}}_{v9c}$	$T_{0}\left({\dot{S}}_{g12}-{\dot{S}}_{g11}+{\dot{S}}_{v10}-{\dot{S}}_{v9c}\right)$
LPEC	${\dot{E}}_{g12}-{\dot{E}}_{g13}$	${\dot{E}}_{v9}-{\dot{E}}_{v8}$	$T_{0}\left({\dot{S}}_{g13}-{\dot{S}}_{g12}+{\dot{S}}_{v9}-{\dot{S}}_{v8}\right)$
**Steam cycle**				
HPST	${\dot{E}}_{v1}-{\dot{E}}_{v2}$	${\dot{W}}_{HPST}$	$T_{0}\left({\dot{S}}_{v2}-{\dot{S}}_{v1}\right)$
IPST	${\dot{E}}_{v4}-{\dot{E}}_{v5b}$	${\dot{W}}_{IPST}$	$T_{0}\left({\dot{S}}_{v5b}-{\dot{S}}_{v4}\right)$
LPST	${\dot{E}}_{v5}-{\dot{E}}_{v6}$	${\dot{W}}_{LPST}$	$T_{0}\left({\dot{S}}_{v6}-{\dot{S}}_{v5}\right)$
LPP	${\dot{W}}_{LPP}$	${\dot{E}}_{v8}-{\dot{E}}_{v7}$	$T_{0}\left({\dot{S}}_{v8}-{\dot{S}}_{v7}\right)$
IPP	${\dot{W}}_{IPP}$	${\dot{E}}_{v11}-{\dot{E}}_{v9a}$	$T_{0}\left({\dot{S}}_{v11}-{\dot{S}}_{v9a}\right)$
HPP	${\dot{W}}_{HPP}$	${\dot{E}}_{v14}-{\dot{E}}_{v9b}$	$T_{0}\left({\dot{S}}_{v14}-{\dot{S}}_{v9b}\right)$
LPD	${\dot{E}}_{v9}+{\dot{E}}_{v10}$	${\dot{E}}_{v10}+{\dot{E}}_{v9a}+{\dot{E}}_{v9b}+{\dot{E}}_{v9c}$	0
IPD	${\dot{E}}_{v13}-{\dot{E}}_{v12}$	${\dot{E}}_{v13}-{\dot{E}}_{v12}$	0
HPD	${\dot{E}}_{v17}-{\dot{E}}_{v16}$	${\dot{E}}_{v17}-{\dot{E}}_{v16}$	0
m	${\dot{E}}_{v5a}+{\dot{E}}_{v5b}$	${\dot{E}}_{v5}$	$T_{0}\left({\dot{S}}_{v5}-{\dot{S}}_{v5b}-{\dot{S}}_{v5a}\right)$
M	${\dot{E}}_{v3a}+{\dot{E}}_{v2}$	${\dot{E}}_{v3}$	$T_{0}\left({\dot{S}}_{v3}-{\dot{S}}_{v3a}-{\dot{S}}_{v2}\right)$
**Electric generator**				
EG	${\dot{W}}_{HPST}+{\dot{W}}_{IPST}+{\dot{W}}_{LPST}$	${\dot{W}}_{LPP}+{\dot{W}}_{IPP}+{\dot{W}}_{HPP}$	0
		$+{\dot{W}}_{CWP}+{\dot{W}}_{mSC}$	

**Table 3 entropy-22-00299-t003:** Exergy flow rates of the resource, product, and irreversibilities of the dissipative components of the combined cycle.

Component	$\dot{F}$	$\dot{P}$	$\dot{R}$	$\dot{I}$
**Gas-side residue**					
Stack	${\dot{E}}_{g13}$		${\dot{E}}_{g13}$	0
**Steam-side residue**					
COND	${\dot{E}}_{v6}+{\dot{W}}_{CWP}$	${\dot{E}}_{v7}$	${\dot{E}}_{{\dot{Q}}_{COND}}$	$T_{0}\left({\dot{S}}_{v7}-{\dot{S}}_{v6}\right)+{\dot{Q}}_{COND}\frac{T_{0}}{T_{COND}}+{\dot{W}}_{CWP}$

**Table 4 entropy-22-00299-t004:** Exergetic costs balances and auxiliary equations for the combined cycle components.

Component	Exergetic Costs Balances	Auxiliary Equations
**Gas turbine cycle**		
External		$E_{f}^{*}={\dot{E}}_{f}$
resources		$E_{g1}^{*}={\dot{E}}_{g1}$
C	$E_{{\dot{W}}_{C}}^{*}+R_{C}^{*}=E_{g2}^{*}-E_{g1}^{*}$
cc	$E_{f}^{*}+R_{cc}^{*}=E_{g3}^{*}-E_{g2}^{*}$	${\dot{E}}_{g4}E_{g3}^{*}={\dot{E}}_{g3}E_{g4}^{*}$
*t*	$E_{g3}^{*}-E_{g4}^{*}+R_{t}^{*}=E_{{\dot{W}}_{mGT}}^{*}+E_{{\dot{W}}_{C}}^{*}$	${\dot{E}}_{{\dot{W}}_{mGT}}E_{{\dot{W}}_{C}}^{*}={\dot{E}}_{{\dot{W}}_{C}}E_{{\dot{W}}_{mGT}}^{*}$
**Heat recovery steam generator**		
HPSH+IPRH	$E_{g4}^{*}-E_{g5}^{*}+R_{HPSH+IPRH}^{*}=E_{v1}^{*}-E_{v17}^{*}+E_{v4}^{*}-E_{v3}^{*}$	${\dot{E}}_{g3}E_{g5}^{*}={\dot{E}}_{g5}E_{g3}^{*}$
HPEV	$E_{g5}^{*}-E_{g6}^{*}+R_{HPEV}^{*}=E_{v17}^{*}-E_{v16}^{*}$	${\dot{E}}_{g3}E_{g6}^{*}={\dot{E}}_{g6}E_{g3}^{*}$
HPEC	$E_{g6}^{*}-E_{g7}^{*}+R_{HPEC}^{*}=E_{v16}^{*}-E_{v15}^{*}$	${\dot{E}}_{g3}E_{g7}^{*}={\dot{E}}_{g7}E_{g3}^{*}$
LPSH	$E_{g7}^{*}-E_{g8}^{*}+R_{LPSH}^{*}=E_{v5a}^{*}-E_{v10}^{*}$	${\dot{E}}_{g3}E_{g8}^{*}={\dot{E}}_{g8}E_{g3}^{*}$
IPSH	$E_{g8}^{*}-E_{g9}^{*}+R_{IPSH}^{*}E^{*}=E_{v3a}^{*}-E_{v13}^{*}$	${\dot{E}}_{g3}E_{g9}^{*}={\dot{E}}_{g9}E_{g3}^{*}$
IPEV	$E_{g9}^{*}-E_{g10}^{*}+R_{IPEV}^{*}=E_{v13}^{*}-E_{v12}^{*}$	${\dot{E}}_{g3}E_{g10}^{*}={\dot{E}}_{g10}E_{g3}^{*}$
HPPH+IPEC	$E_{g10}^{*}-E_{g11}^{*}+R_{HPPH+IPEC}^{*}=E_{v12}^{*}-E_{v11}^{*}+E_{v15}^{*}-E_{v14}^{*}$	${\dot{E}}_{g3}E_{g11}^{*}={\dot{E}}_{g11}E_{g3}^{*}$
LPEV	$E_{g11}^{*}-E_{g12}^{*}+R_{LPEV}^{*}=E_{v10}^{*}-E_{v9c}^{*}$	${\dot{E}}_{g3}E_{g12}^{*}={\dot{E}}_{g12}E_{g3}^{*}$
LPEC	$E_{g12}^{*}-E_{g13}^{*}+R_{LPEC}^{*}=E_{v9}^{*}-E_{v8}^{*}$	${\dot{E}}_{g3}E_{g13}^{*}={\dot{E}}_{g13}E_{g3}^{*}$
**Steam cycle**		
HPST	$E_{{\dot{W}}_{HPST}}^{*}=E_{v1}^{*}-E_{v2}^{*}+R_{HPST}^{*}$	${\dot{E}}_{v1}E_{v2}^{*}={\dot{E}}_{v2}E_{v1}^{*}$
IPST	$E_{{\dot{W}}_{IPST}}^{*}=E_{v4}^{*}-E_{v5b}^{*}+R_{IPST}^{*}$	${\dot{E}}_{v4}E_{v5b}^{*}={\dot{E}}_{v5b}E_{v4}^{*}$
LPST	$E_{{\dot{W}}_{LPST}}^{*}=E_{v5}^{*}-E_{v6}^{*}+R_{LPST}^{*}$	${\dot{E}}_{v5}E_{v6}^{*}={\dot{E}}_{v6}E_{v5}^{*}$
LPP	$E_{{\dot{W}}_{LPP}}^{*}+R_{LPP}^{*}=E_{v8}^{*}-E_{v7}^{*}$
IPP	$E_{{\dot{W}}_{IPP}}^{*}+R_{IPP}^{*}=E_{v11}^{*}-E_{v9a}^{*}$
HPP	$E_{{\dot{W}}_{HPP}}^{*}+R_{HPP}^{*}=E_{v14}^{*}-E_{v9b}^{*}$
LPD	$E_{v9}^{*}=E_{v9a}^{*}+E_{v9b}^{*}+E_{v9c}^{*}$	$E_{v9a}^{*}{\dot{E}}_{v9c}=E_{v9c}^{*}{\dot{E}}_{v9a}$
		$E_{v9a}^{*}{\dot{E}}_{v9b}=E_{v9b}^{*}{\dot{E}}_{v9a}$
m	$E_{v5a}^{*}+E_{v5b}^{*}=E_{v5}^{*}$
M	$E_{v2}^{*}+R_{M}^{*}=E_{v3}^{*}-E_{v3a}^{*}$
COND	$E_{{\dot{Q}}_{COND}}^{*}=E_{v6}^{*}-E_{v7}^{*}+{\dot{E}}_{{\dot{W}}_{CWP}}^{*}$	$E_{v7}^{*}={\dot{E}}_{v7}$
EG		${\dot{E}}_{{\dot{W}}_{HPP}}E_{{\dot{W}}_{mSC}}^{*}={\dot{E}}_{{\dot{W}}_{mSC}}E_{{\dot{W}}_{HPP}}^{*}$
	$E_{{\dot{W}}_{HPST}}^{*}+E_{{\dot{W}}_{IPST}}^{*}+E_{{\dot{W}}_{LPST}}^{*}$	${\dot{E}}_{{\dot{W}}_{IPP}}E_{{\dot{W}}_{mSC}}^{*}={\dot{E}}_{{\dot{W}}_{mSC}}E_{{\dot{W}}_{IPP}}^{*}$
	$=E_{{\dot{W}}_{HPP}}^{*}+E_{{\dot{W}}_{IPP}}^{*}+E_{{\dot{W}}_{LPP}}^{*}+{\dot{E}}_{{\dot{W}}_{CWP}}^{*}+E_{{\dot{W}}_{mSC}}^{*}$	${\dot{E}}_{{\dot{W}}_{LPP}}E_{{\dot{W}}_{mSC}}^{*}={\dot{E}}_{{\dot{W}}_{mSC}}E_{{\dot{W}}_{LPP}}^{*}$
		${\dot{E}}_{{\dot{W}}_{CWP}}E_{{\dot{W}}_{mSC}}^{*}={\dot{E}}_{{\dot{W}}_{mSC}}E_{{\dot{W}}_{CWP}}^{*}$

**Table 5 entropy-22-00299-t005:** Thermodynamic states of the combined cycle at actual operating condition.

Thermodynamic States											
State	*T*	*P*	*v*	*s*	*h*	ε	*X*	$\dot{m}$	$\dot{E}$	$\dot{H}$	$\dot{S}$
	(°C)	(bar)	$\left(\frac{m^{3}}{\mathrm{kg}}\right)$	$\left(\frac{\mathrm{kJ}}{\mathrm{kg}\;K}\right)$	$\left(\frac{\mathrm{kJ}}{\mathrm{kg}}\right)$	$\left(\frac{\mathrm{kJ}}{\mathrm{kg}}\right)$	(-)	$\left(\frac{\mathrm{kg}}{s}\right)$	(MW)	(MW)	$\left(\frac{\mathrm{MW}}{K}\right)$
**Gas turbine cycle**											
*f*	25.00	16.21		9.37		43,275.51		15.36	664.79	664.79	0.14
g1	25.00	1.01	0.853	6.76	0.00	0.00		626.82	0.00	0.00	4.24
g2	433.77	16.21	0.126	6.90	445.37	404.49		626.82	253.54	279.17	4.32
g3	1300.00	15.89	0.292	7.80	1618.70	1231.00		642.18	790.52	1039.50	5.01
g4	617.58	1.02	2.565	7.88	750.46	338.09		642.18	217.11	481.93	5.06
${\dot{W}}_{C}$									279.17		
${\dot{W}}_{mGT}$									278.40		
g5	489.28	1.02	2.200	7.71	597.33	237.89		642.18	152.77	383.59	4.95
g6	374.28	1.02	1.870	7.52	460.07	156.19		642.18	100.30	295.45	4.83
g7	299.46	1.02	1.655	7.39	376.39	110.11		642.18	70.71	241.71	4.75
g8	292.51	1.02	1.636	7.38	368.62	106.00		642.18	68.07	236.72	4.74
**Heat recovery steam generator**											
g9	290.50	1.02	1.632	7.38	366.37	104.76		642.18	67.28	235.28	4.74
g10	262.57	1.02	1.553	7.33	335.14	88.99		642.18	57.15	215.22	4.70
g11	231.48	1.02	1.465	7.27	301.18	72.49		642.18	46.55	193.41	4.67
g12	164.16	1.01	1.271	7.13	227.66	40.92		642.18	26.28	146.20	4.58
g13	98.62	1.01	1.081	6.97	156.82	16.68		642.18	10.71	100.71	4.48
**Steam cycle**											
v1	525.80	127.38	0.026	6.54	3410.26	1464.34	SW	76.78	112.43	261.84	0.50
v2	328.25	32.06	0.081	6.62	3058.97	1088.92	SW	76.78	83.61	234.86	0.51
v3	321.60	32.06	0.079	6.60	3042.57	1080.69	SW	88.06	95.17	267.94	0.58
v3a	278.75	32.06	0.071	6.40	2930.98	1027.20	SW	11.29	11.59	33.08	0.07
v4	525.80	32.06	0.113	7.28	3512.97	1347.96	SW	88.06	118.71	309.37	0.64
v5a	246.67	3.53	0.671	7.43	2959.35	749.43	SW	21.99	16.48	65.08	0.16
v5b	246.67	3.53	0.671	7.43	2959.35	749.43	SW	88.06	66.00	260.62	0.65
v5	246.67	3.53	0.671	7.43	2959.35	749.43	SW	110.06	82.48	325.70	0.82
v6	41.03	0.08	17.167	7.67	2397.81	115.22	0.926	110.06	12.68	263.90	0.84
v7	41.03	0.08	0.001	0.59	171.85	1.65	0	110.06	0.18	18.91	0.06
v8	41.07	3.53	0.001	0.59	172.24	1.94	CLW	110.06	0.21	18.96	0.06
v9	139.16	3.53	0.001	1.73	585.60	74.19	0	110.06	8.16	64.45	0.19
v9a	139.16	3.53	0.001	1.73	585.60	74.19	0	11.29	0.84	6.61	0.02
v9b	139.16	3.53	0.001	1.73	585.60	74.19	0	76.78	5.70	44.96	0.13
v9c	139.16	3.53	0.001	1.73	585.60	74.19	0	21.99	1.63	12.88	0.04
v10	139.16	3.53	0.520	6.94	2732.36	668.58	1	21.99	14.70	60.09	0.15
v11	139.59	32.06	0.001	1.73	589.24	77.37	CLW	11.29	0.87	6.65	0.02
v12	237.57	32.06	0.001	2.68	1025.95	231.57	0	11.29	2.61	11.58	0.03
v13	237.57	32.06	0.062	6.16	2803.23	971.30	1	11.29	10.96	31.64	0.07
v14	140.38	127.38	0.001	1.74	601.28	88.14	CLW	76.78	6.77	46.17	0.13
v15	191.85	127.38	0.001	2.24	821.10	158.76	CLW	76.78	12.19	63.04	0.17
v16	329.28	127.38	0.002	3.54	1520.98	468.90	0	76.78	36.00	116.78	0.27
v17	329.28	127.38	0.013	5.45	2669.03	1048.77	1	76.78	80.52	204.92	0.42
${\dot{W}}_{LPP}$									0.04		
${\dot{W}}_{IPP}$									0.04		
${\dot{W}}_{HPP}$									1.20		
${\dot{W}}_{HPST}$									26.97		
${\dot{W}}_{IPST}$									48.75		
${\dot{W}}_{LPST}$									61.80		
${\dot{W}}_{CWP}$									1.74		
${\dot{W}}_{mSC}$									134.48		
${\dot{Q}}_{COND}$									12.50		

**Table 6 entropy-22-00299-t006:** Thermodynamic performance indicators of the combined cycle.

**Gas Turbine**									
	${\dot{m}}_{air}$	${\dot{m}}_{f}$	${\dot{m}}_{cg}$	$w_{mGT}$	$q_{HGT}$	$SFC_{GT}$	$\eta_{thGT}$	$\eta_{exGT}$
	(kg${}_{\mathrm{air}}/s$)	(kg${}_{f}/s$)	(kg${}_{cg}$/s)	(kJ/kg${}_{\mathrm{air}}$)	(kJ/kg${}_{\mathrm{air}}$)	(kg${}_{f}$/kWh)	(%)	(%)
1GT	313.40	7.68	321.08	444.14	1,262.87	0.1986	35.16	40.25	
2GT	626.81	15.36	642.17	444.14	1,262.87	0.1986	35.16	40.25	
**HRSG**		**Steam cycle**							
${\dot{Q}}_{H_{HSRG}}$		${\dot{m}}_{HP}$	${\dot{m}}_{IP}$	${\dot{m}}_{LP}$	$w_{mSC}$	${\dot{W}}_{mSC}$	$SSC_{ST}$	$\eta_{thST}$	$\eta_{exST}$
(MW)		(kg${}_{s}$/s)	(kg${}_{s}$/s)	(kJ/kg${}_{s}$)	(kJ/kg${}_{s}$)	(MW)	(kg${}_{s}$/kWh)	(%)	(%)
387.00		76.77	11.28	21.99	1,751.65	134.48	2.05	34.75	82.65
**Combined cycle with a 2×2×1 arrangement**									
		${\dot{m}}_{f}$	${\dot{W}}_{mCC}$	$SSC_{CC}$	$SFC_{CC}$	$\eta_{thCC}$	$\eta_{exCC}$
		(kg${}_{f}/s$)	(MW)	(kg${}_{s}$/kWh)	(kg${}_{f}$/kWh)	(%)	(%)
		15.36	412.88	0.67	0.13	54.30	62.15		

**Table 7 entropy-22-00299-t007:** Fuel ($\dot{F}$), product ($\dot{P}$), residue ($\dot{R}$), irreversibility ($\dot{I}$), entropy change ($T_{0}\Delta \dot{S}$), and exergetic efficiency ($\eta_{ex}$) of the components in the gas and steam sides, and irreversibility and exergetic efficiency of the component in the combined cycle.

Component	Gas Side							Steam Side							Combined Cycle	
	$\dot{F}$	$\dot{P}$	$\dot{R}$	$\dot{I}$	$T_{0}\Delta \dot{S}$	$\eta_{\mathrm{ex}}$		$\dot{F}$	$\dot{P}$	$\dot{R}$	$\dot{I}$	$T_{0}\Delta \dot{S}$	$\eta_{\mathrm{ex}}$		$\dot{I}$	$\eta_{\mathrm{ex}}$
	(MW)	(MW)	(MW)	(MW)	(MW)	(%)		(MW)	(MW)	(MW)	(MW)	(MW)	(%)		(MW)	(%)
**Gas turbine cycle**																
C	279.17	253.54		25.62	25.62	90.82									25.62	90.82
cc	664.79	536.98		127.81	223.35	80.77									127.81	80.77
t	573.41	557.57		15.84	15.84	97.24									15.84	97.24
GT	664.79	495.51		169.28	264.82	74.54									169.28	74.54
**Heat recovery steam generator**																
HPSH+IPRH	64.35	62.65		1.69	−33.99	97.37		62.65	55.44		7.21	42.89	88.50		8.90	86.17
HPEV	52.47	50.61		1.85	−35.68	96.47		50.61	44.52		6.09	43.62	87.96		7.95	84.86
HPEC	29.59	27.37		2.21	−24.15	92.52		27.37	23.81		3.56	29.92	86.99		5.77	80.48
LPSH	2.64	2.38		0.26	−2.35	89.97		2.38	1.78		0.60	3.21	74.81		0.86	67.31
IPSH	0.79	0.68		0.11	−0.65	85.76		0.68	0.63		0.05	0.81	92.71		0.16	79.51
IPEV	10.13	9.17		0.96	−9.93	90.52		9.17	8.35		0.82	11.71	91.03		1.78	82.40
HPPH+IPEC	10.59	9.30		1.29	−11.22	87.77		9.30	7.16		2.13	14.64	77.05		3.43	67.63
LPEV	20.28	17.17		3.11	−26.93	84.67		17.17	13.07		4.10	34.14	76.13		7.21	64.46
LPEC	15.56	11.74		3.82	−29.93	75.44		11.74	7.95		3.79	37.54	67.71		7.61	51.09
HRSG	217.11	191.08		15.32	−174.82	88.01		191.08	162.72		28.36	218.50	85.16		85.16	74.95
HRSG${}_{g}$+GT	664.79	480.19		184.60	90.00	72.23										
Stack	10.71	0.00	10.71	0.00	0											
**Steam cycle**																
HPST								28.82	26.97		1.85	1.85	93.57		1.85	93.57
IPST								52.71	48.75		3.96	3.96	92.50		3.96	92.50
LPST								69.80	61.80		8.00	8.00	88.54		8.00	88.54
LPP								0.04	0.03		0.01	0.01	74.74		0.01	74.74
IPP								0.04	0.04		0.01	0.01	87.44		0.01	87.44
HPP								1.20	1.07		0.13	0.13	88.99		0.13	88.99
LPD								22.87	22.87		0.00	0.00	100.00		0.00	100.00
IPD								8.35	8.35		0.00	0.00	100.00		0.00	100.00
HPD								44.52	44.52		0.00	0.00	100.00		0.00	100.00
m								82.48	82.48		0.00	0.00	100.00		0.00	100.00
M								95.20	95.17		0.03	0.03	99.97		0.03	99.97
SC${}_{prod}$								406.04	392.06		13.98	13.98	96.56		13.98	96.56
HRSG${}_{v}$+SC${}_{prod}$											42.34	232.48	96.56		99.14	96.56
COND								14.25		12.50	1.75	−232.48			1.75	

**Table 8 entropy-22-00299-t008:** Residue cost allocation ratios of the productive components.

Productive	Cost Allocation Ratios for the Exhaust				Cost Allocation Ratios for the Waste			
	Gases Dissipated in the Stack, μ (-)				Heat Dissipated in the Condenser, β (-)			
Component	Criterion							
	C1	C2	C3	C4	C1	C2	C3	C4
**Gas turbine cycle**								
C	0.191	0.139	0.321	0.262				
cc	1.992	0.693	0.679	0.738				
t	0.118	0.086						
**Heat recovery steam generator**								
HPSH+IPRH	−0.253	0.009			0.184	0.172	0.337	0.252
HPEV	−0.265	0.010			0.188	0.144	0.271	0.227
HPEC	−0.180	0.012			0.129	0.084	0.145	0.140
LPSH	−0.017	0.001			0.014	0.014	0.011	0.013
IPSH	−0.005	0.001			0.003	0.001	0.004	0.004
IPEV	−0.074	0.005			0.050	0.019	0.051	0.052
HPPH+IPEC	−0.083	0.007			0.063	0.050	0.044	0.058
LPEV	−0.200	0.017			0.147	0.097	0.079	0.125
LPEC	−0.223	0.021			0.161	0.089	0.048	0.122
**Steam cycle**								
HPST					0.008	0.044		
IPST					0.017	0.093		
LPST					0.034	0.188		
LPP					0.000	0.000255	0.000195	0.000107
IPP					0.000	0.000122	0.000218	0.000101
HPP					0.001	0.00312	0.006514	0.002955
LPD					0.00	0.00		
IPD					0.00	0.00		
HPD					0.00	0.00		
m					0.00	0.00		
M					0.000119	0.000654	0.00367	0.00326
Total	1.000	1.000	1.000	1.000	1.000	1.000	1.000	1.000

**Table 9 entropy-22-00299-t009:** Exergetic costs of the combined cycle streams for different residue cost allocation criteria.

Stream	Exergetic Costs,${\dot{E}}^{*}$(MW)			
	Criterion			
	C1	C2	C3	C4
**Gas turbine cycle**				
*f*	664.789	664.789	664.789	664.789
g1	0.000	0.000	0.000	0.000
g2	401.823	388.937	392.003	391.159
g3	1096.199	1063.710	1066.608	1066.608
g4	301.064	292.141	292.937	292.937
${\dot{W}}_{C}$	398.991	386.935	387.368	387.368
${\dot{W}}_{mTG}$	397.894	385.871	386.303	386.303
**Heat recovery steam generator (gas side)**				
g5	211.838	205.559	206.119	206.119
g6	139.083	134.961	135.328	135.328
g7	98.056	95.150	95.409	95.409
g8	94.393	91.595	91.845	91.845
g9	93.293	90.528	90.774	90.774
g10	79.243	76.894	77.104	77.104
g11	64.556	62.643	62.813	62.813
g12	36.436	35.356	35.452	35.452
g13	14.852	14.412	14.451	14.451
**Heat recovery steam generator (steam side)**				
**Steam cycle**				
v1	191.648	194.021	201.492	200.559
v2	142.514	144.279	149.835	149.141
v3	171.194	174.535	180.021	179.647
v3a	28.677	30.238	30.083	30.414
v4	208.322	211.659	219.759	218.362
v5a	37.434	38.942	37.982	39.775
v5b	115.822	117.677	122.181	121.404
v5	153.256	156.619	160.163	161.179
v6	23.562	24.079	24.624	24.780
v7	0.181	0.181	0.181	0.181
v8	0.267	0.277	0.275	0.273
v9	22.880	23.978	22.633	24.726
v9a	2.346	2.459	2.321	2.536
v9b	15.962	16.728	15.789	17.249
v9c	4.572	4.791	4.523	4.941
v10	33.660	34.979	34.115	35.840
v11	2.428	2.547	2.412	2.623
v12	13.249	14.887	13.810	14.090
v13	27.555	29.129	28.905	29.238
v14	18.366	19.310	18.465	19.826
v15	22.684	22.709	22.580	24.273
v16	64.499	65.002	66.563	68.137
v17	138.351	139.697	144.952	145.348
${\dot{W}}_{LPP}$	0.085	0.089	0.089	0.089
${\dot{W}}_{IPP}$	0.082	0.085	0.085	0.085
${\dot{W}}_{HPP}$	2.389	2.497	2.493	2.493
${\dot{W}}_{HPST}$	49.348	50.944	51.658	51.419
${\dot{W}}_{IPST}$	92.957	96.547	97.579	96.958
${\dot{W}}_{LPST}$	130.618	137.727	135.539	136.399
${\dot{W}}_{mST}$	266.894	278.918	278.486	278.486
${\dot{Q}}_{COND}$	3.473	3.629	3.623	3.623
${\dot{W}}_{mTOT}$	26.853	27.527	28.066	28.222

**Table 10 entropy-22-00299-t010:** Exergetic cost balances for the combined cycle components.

Criterion	C1		C2		C3		C4
Component	${\dot{F}}^{*}$	${\dot{P}}^{*}$	${{\dot{E}}_{g13}}^{*}$	${\dot{E}}_{{\dot{Q}}_{\mathrm{COND}}}^{*}$		${\dot{F}}^{*}$	${\dot{P}}^{*}$	${{\dot{E}}_{g13}}^{*}$	${\dot{E}}_{{\dot{Q}}_{\mathrm{COND}}}^{*}$		${\dot{F}}^{*}$	${\dot{P}}^{*}$	${{\dot{E}}_{g13}}^{*}$	${\dot{E}}_{{\dot{Q}}_{\mathrm{COND}}}^{*}$		${\dot{F}}^{*}$	${\dot{P}}^{*}$	${{\dot{E}}_{g13}}^{*}$	${\dot{E}}_{{\dot{Q}}_{\mathrm{COND}}}^{*}$
(MW)		(MW)		(MW)		(MW)
**Gas turbine**
C	398.99	401.82	2.83			386.935	388.937	2.002			387.368	392.003	4.635			387.368	391.159	3.791	
cc	664.79	694.38	29.59			664.789	674.773	9.984			664.789	674.605	9.816			664.789	675.449	10.660	
t	795.13	796.89	1.75			771.569	772.806	1.238			773.671	773.671	0.000			773.671	773.671	0.000	
**Heat recovery steam generator**
HPSH+IPRH	89.23	90.42	−3.76	4.95		86.582	91.448	0.124	4.743		86.818	96.279	0.000	9.462		86.818	93.926	0	7.109
HPEV	72.76	73.85	−3.94	5.04		70.599	74.695	0.145	3.952		70.791	78.389	0.000	7.598		70.791	77.211	0	6.420
HPEC	41.03	41.81	−2.67	3.46		39.811	42.293	0.173	2.309		39.919	43.983	0.000	4.064		39.919	43.864	0	3.944
LPSH	3.66	3.77	−0.26	0.37		3.555	3.964	0.021	0.388		3.564	3.868	0.000	0.303		3.564	3.935	0	0.370
IPSH	1.10	1.12	−0.07	0.09		1.068	1.109	0.009	0.032		1.071	1.178	0.000	0.108		1.071	1.177	0	0.106
IPEV	14.05	14.31	−1.10	1.35		13.633	14.242	0.075	0.534		13.670	15.095	0.000	1.425		13.670	15.148	0	1.477
HPPH+IPEC	14.69	15.14	−1.24	1.69		14.252	15.739	0.101	1.386		14.290	15.513	0.000	1.222		14.290	15.914	0	1.624
LPEV	28.12	29.09	−2.98	3.94		27.287	30.188	0.243	2.658		27.361	29.592	0.000	2.231		27.361	30.899	0	3.538
LPEC	21.58	22.61	−3.31	4.34		20.944	23.701	0.299	2.458		21.001	22.358	0.000	1.357		21.001	24.453	0	3.452
HRSG	286.21	292.13	−19.32	25.24		277.729	297.377	1.188	18.460		278.486	306.254	0.000	27.769		278.486	306.527	0	28.041
Stack	14.85		14.85			14.412		14.412			14.451		14.451			14.451		14.451	
**Steam cycle**
HPST	49.13	49.35		0.21		49.742	50.944		1.202		51.658	51.658		0.000		51.419	51.419		0.000
IPST	92.50	92.96		0.46		93.982	96.547		2.565		97.579	97.579		0.000		96.958	96.958		0.000
LPST	129.69	130.62		0.92		132.540	137.727		5.186		135.539	135.539		0.000		136.399	136.399		0.000
LPP	0.09	0.09		0.00		0.089	0.096		0.007		0.089	0.094		0.005		0.089	0.092		0.003
IPP	0.08	0.08		0.00		0.085	0.089		0.003		0.085	0.091		0.006		0.085	0.088		0.003
HPP	2.39	2.40		0.02		2.497	2.582		0.086		2.493	2.676		0.183		2.493	2.576		0.083
LPD	56.54	56.54		0.00		58.957	58.957		0.000		56.748	56.748		0.000		60.566	60.566		0.000
IPD	14.31	14.31		0.00		14.242	14.242		0.000		15.095	15.095		0.000		15.148	15.148		0.000
HPD	73.85	73.85		0.00		74.695	74.695		0.000		78.389	78.389		0.000		77.211	77.211		0.000
m	153.26	153.26		0.00		156.619	156.619		0.000		160.163	160.163		0.000		161.179	161.179		0.000
M	171.19	171.19		0.00		174.517	174.535		0.018		179.918	180.021		0.103		179.555	179.647		0.092
COND	26.85			26.85		27.527			27.527		28.066			28.066		28.222			28.222
EG	272.92	272.92				285.217	285.217				284.776	284.776				284.776	284.776

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
