# Peer review of "An Irreversibility-Based Criterion to Determine the Cost Formation of Residues in a Three-Pressure-Level Combined Cycle"

_entropy, 2020, doi:10.3390/e22030299_

Round 1

Reviewer 1 Report

The paper you are intending to publish in Entropy falls within the scope of the Journal, however, it does not provide a novelty in relation to what is found in current literature on exergoeconomics. Besides the paper is too long to read and contains many information taken from other previous works.

Furthermore, the following points should be improved:

1) The Abstract is not objective, it should be rewritten and focus mainly on what is your novelty or proposal as well as the results obtained.

2) It is not necessary to write a long introduction, it is preferable to have a more objective introduction that draws the reader´s attention.

3) Authors should eliminate parts which have been published before and communicate the most important part of the research. Remember that this is a paper, not a technical report. For example, explication of exergy cost theory.

4) How did the authors validate the results of the thermodynamic model?, did you perform a statistical model which provided the percentage of errors or deviations from a reference model?.

5) Which reference state did you consider for your power plant for making the calculations?, was it the design point or a point after an overhaul or an arbitrary state?

6) In the conclusions, the authors commented that the heat dissipated from the condenser is formed in the all productive components of the heat recovery steam generator and those of the steam cycle. This is something already known, this is not a finding of their results.

7) What is the physical interpretation of negative exergy costs as those shown in Table 10 in criterion C1. Is that possible?

8) The numbering of the references is incorrect. It should be corrected.

9) An approach similar to the one proposed by the authors can be reviewed in:

Alejandro Zaleta-Aguilar, Abraham Olivares-Arriaga, Sergio Cano-Andrade, David A. Rodriguez-Alejandro,
β-characterization by irreversibility analysis: A thermoeconomic diagnosis method,Energy,Volume 111,2016.

https://doi.org/10.1016/j.energy.2016.06.012.

J.J. Pacheco Ibarra, V.H. Rangel Hernández, A. Zaleta Aguilar, A. Valero,
Hybrid Fuel Impact Reconciliation Method: An integral tool for thermoeconomic diagnosis. Energy. Volume 35, Issue 5,
2010. https://doi.org/10.1016/j.energy.2010.01.026.

9) Finally, according to thermoeconomic theories, this type of analysis should be applied to systems that have, at least, two different products with different energy quality, so it makes sense to apply them. For example:

Rangel-Hernandez, V.H.; Torres, C.; Zaleta-Aguilar, A.; Gomez-Martinez, M.A. The Exergy Costs of Electrical Power, Cooling, and Waste Heat from a Hybrid System Based on a Solid Oxide Fuel Cell and an Absorption Refrigeration System. Energies 2019, 12, 3476.

V.H. Rangel-Hernández, J.J. Ramírez-Minguela, Ludger Blum, A.M. Niño-Avendaño, R. Ornelas. Parametric analysis of the exergoeconomic variables of a solid oxide fuel cell (SOFC) coupled with a vapour-adsorption refrigeration system (VARS). Energy Conversion and Management. Volume 172. 2018. https://doi.org/10.1016/j.enconman.2018.07.040.

A. Zaleta-Aguilar, D.A. Rodriguez-Alejandro, V.H. Rangel-Hernández,
Application of an exergy-based thermo characterization approach to diagnose the operation of a biomass-fueled gasifier,
Biomass and Bioenergy. Volume 116,
2018. https://doi.org/10.1016/j.biombioe.2018.05.008.

Author Response

Reviewer #1: ("Please see the attachment." )

Thank you for these observations:

The paper you are intending to publish in Entropy falls within the scope of the Journal, however, it does not provide a novelty in relation to what is found in current literature on exergoeconomics.

Point 1: Besides the paper is too long to read and contains many information taken from other previous works.

Response 1: Based on your observations, the number of pages of the article was reduced.

Point 2: 1) The Abstract is not objective, it should be rewritten and focus mainly on what is your novelty or proposal as well as the results obtained.

Response 2: The abstract has been rewritten and focus mainly on what is your novelty or proposal as well as the obtained results.

Point 3: 2) It is not necessary to write a long introduction, it is preferable to have a more objective introduction that draws the reader´s attention.

Response 3: The introduction has been rewritten to have a more objective introduction that draws the reader´s attention.

Point 4: 3) Authors should eliminate parts which have been published before and communicate the most important part of the research. Remember that this is a paper, not a technical report. For example, explication of exergy cost theory.

Response 4: We have removed the parts that have been published before and now we focus the article on the most important of the research.

Point 5: 4) How did the authors validate the results of the thermodynamic model? Did you perform a statistical model which provided the percentage of errors or deviations from a reference model?

Response 5:

For the first question, the results of the thermodynamic model were validated with the design thermal balance obtained from the record of the combined cycle plant. For the second question, no statistical model was used but was based on a thermal balance obtained from a combined cycle plant in operation.

Point 6: 5) Which reference state did you consider for your power plant for making the calculations? Was it the design point or a point after an overhaul or an arbitrary state?

 Response 6: The design condition is the reference state information, which contained within the thermal balance of the combined cycle reported by the plant.

Point 7: 6) In the conclusions, the authors commented that the heat dissipated from the condenser is formed in the all productive components of the heat recovery steam generator and those of the steam cycle. This is something already known, this is not a finding of their results.

Response 7: The components involved in the formation of the heat discarded by the condenser depend on the chosen cost allocation criterion. For the distributed entropy and exergy criteria, the components of the heat recovery steam generator, the water pumps (high, intermediate and low pressure) and the mixer M are the only components participating in the formation of this residue. Meanwhile, in the case of the generated entropy, all the components participate in its formation. Thus, with the proposed criteria, being an extension of the generated entropy criterion, it turns out that all the components form this residue.

Point 8: 7) What is the physical interpretation of negative exergy costs as those shown in Table 10 in criterion C1. Is that possible?

Response 8: For criterion C1 corresponding to the entropy changes, the allocation costs for the components of the heat recovery generator are due to the decrease in entropy presented in these components, that is, these components favor the decrease of the cost of residue formation.

Point 9: 8) The numbering of the references is incorrect. It should be corrected.

Response 9: This mistake has already been corrected.

Point 10: 9) An approach similar to the one proposed by the authors can be reviewed in:

Alejandro Zaleta-Aguilar, Abraham Olivares-Arriaga, Sergio Cano-Andrade, David A. Rodriguez-Alejandro,

β-characterization by irreversibility analysis: A thermoeconomic diagnosis method, Energy,Volume 111,2016.

https://doi.org/10.1016/j.energy.2016.06.012.

J.J. Pacheco Ibarra, V.H. Rangel Hernández, A. Zaleta Aguilar, A. Valero,

Hybrid Fuel Impact Reconciliation Method: An integral tool for thermoeconomic diagnosis. Energy. Volume 35, Issue 5,

  1. https://doi.org/10.1016/j.energy.2010.01.026.

Response 10: These two articles refer to thermoeconomic diagnoses to energy systems (comparison between two operating conditions of the system, for example, actual condition with respect to design condition, actual condition vs. condition after an overhault, etc.), which is based on the analysis of malfunctions-dysfunctions. In one hand, the first work (Zaleta et al.) pursues the symbolic thermoeconomic theory of VALERO et al., from which the formula of the impact on fuel at different loads is derived. The second work also performs a malfunction analysis based on the quantification of the irreversibilities of the system components. However, none of these works considers the process of residue formation. On the other hand, this work evaluates the impact of the residues formation process in the production costs of a combined cycle plant at only one operating condition. However, it is working on an upcoming malfunction analysis work based on the irreversibility criteria.

Point 11: 9) Finally, according to thermoeconomic theories, this type of analysis should be applied to systems that have, at least, two different products with different energy quality, so it makes sense to apply them. For example:

Rangel-Hernandez, V.H.; Torres, C.; Zaleta-Aguilar, A.; Gomez-Martinez, M.A. The Exergy Costs of Electrical Power, Cooling, and Waste Heat from a Hybrid System Based on a Solid Oxide Fuel Cell and an Absorption Refrigeration System. Energies 2019, 12, 3476.

V.H. Rangel-Hernández, J.J. Ramírez-Minguela, Ludger Blum, A.M. Niño-Avendaño, R. Ornelas. Parametric analysis of the exergoeconomic variables of a solid oxide fuel cell (SOFC) coupled with a vapour-adsorption refrigeration system (VARS). Energy Conversion and Management. Volume 172. 2018. https://doi.org/10.1016/j.enconman.2018.07.040.

  1. Zaleta-Aguilar, D.A. Rodriguez-Alejandro, V.H. Rangel-Hernández,

Application of an exergy-based thermo characterization approach to diagnose the operation of a biomass-fueled gasifier,

Biomass and Bioenergy. Volume 116,

  1. https://doi.org/10.1016/j.biombioe.2018.05.008.

Response 11: The criteria of irreversibilities proposed in our article is considered as a future analysis of the systems studied in the mentioned articles.

Reviewer 2 Report

The paper “An irreversibility-based criterion to determine the cost formation of residues in a three-pressure-level combined cycle” proposed a criterion for waste cost allocation in Thermoeconomics.

The paper is too long and I suggest to shorten it by eliminating concepts widely known in literature.

Check reference numbering. Something went wrong, since it started with [39] (see line 14); I suggest to avoid using “we” or “you” when explaining or introducing concepts; “Exergy analysis” is more common than “exergetic analysis”; Lines 1-2: “…from conventional energetic and economic analyses”. In my opinion, from conventional exergy analysis as well. Line 110: “First” instead of “fisrt”; Line 125: the following sentence “The results of the proposed criterion are in agreement with the compared criteria” should be reported in an abstract and not within the introduction. Please consider the possibility to erase it; In section 2, the detailed modelling of the investigated case study is presented; in order to shorten this section, some equations could be avoided or organized in a dedicated appendix; Line 334: the compared criteria for waste cost allocation are presented. Please put reference for the three criteria proposed in literature; Eq (41): why is the flow Ev7 is considered as an external resource? Values reported in Table 4 are not results of simulations but description of case study. It could be placed on section 2.1; Line 355: which software did you use? Table 7 and Figure 7 provide the same information. I suggest to avoid repeated information, if possible. This advice is also extended for other tables and figures such as Fig 8 and Table 9, or Fig 9 and table 10. in general, method is introduced before the case study. After that, all equations are applied for the case study. You might consider this suggestion. See for instance http://dx.doi.org/10.1016/j.energy.2012.07.034

Author Response

Reviewer #2: ("Please see the attachment.")

Thank you for these observations:

The paper “An irreversibility-based criterion to determine the cost formation of residues in a three-pressure-level combined cycle” proposed a criterion for waste cost allocation in Thermoeconomics.

Point 1: The paper is too long and I suggest to shorten it by eliminating concepts widely known in literature.

Response 1: Based on your observations, the number of pages of the article was reduced.

Point 2: Check reference numbering. Something went wrong, since it started with [39] (see line 14);

Response 2: This mistake has already been corrected.

Point 3: I suggest to avoid using “we” or “you” when explaining or introducing concepts;

Response 3: This suggestion has been taken into account in this article.

Point 4: “Exergy analysis” is more common than “exergetic analysis”;

Response 4: Your suggestion has been taken into account in this article.

Point 5: Lines 1-2: “…from conventional energetic and economic analyses”. In my opinion, from conventional exergy analysis as well.

Response 5: This suggestion has been taken into account in this article.

Point 6: Line 110: “First” instead of “fisrt”;

Response 6: This mistake has already been corrected.

Point 7: Line 125: the following sentence “The results of the proposed criterion are in agreement with the compared criteria” should be reported in an abstract and not within the introduction. Please consider the possibility to erase it;

Response 7: Based on your observation, this sentence has been removed and placed in the abstract.

Point 8: In section 2, the detailed modelling of the investigated case study is presented; in order to shorten this section, some equations could be avoided or organized in a dedicated appendix;

Response 8: In section 2, the detailed model of study case is presented, because for our study it is important to know the level of aggregation of the study system, as well as the equations that govern its energy streams.

Point 9: Line 334: the compared criteria for waste cost allocation are presented. Please put reference for the three criteria proposed in literature;

Response 9: The references have already been put for the three criteria proposed in the literature.

Point 10: Eq (41): why is the flow Ev7 is considered as an external resource?

Response 10: This stream is an external resource because it is considered as the starting point of the steam cycle, as well as the point where water losses are replenished.

Point 11: Values reported in Table 4 are not results of simulations but description of case study. It could be placed on section 2.1;

Response 11: Based on your observation, the Table 4 was sent to section 4.1.

Point 12: Line 355: which software did you use?

Response 12: No software was used, we developed all the models and the numerical tools.

Point 13: Table 7 and Figure 7 provide the same information. I suggest to avoid repeated information, if possible. This advice is also extended for other tables and figures such as Fig 8 and Table 9, or Fig 9 and table 10.

Response 13: Considering this situation, Figure 7 was removed and Table 7 remained. Besides, Figure 8 was removed and Table 9 remained. On the other hand, Figure 9 and Table 10 complement the information.

Point 14: In general, method is introduced before the case study. After that, all equations are applied for the case study. You might consider this suggestion. See for instance http://dx.doi.org/10.1016/j.energy.2012.07.034.

Response 14: With this article the ideas of this work has been reinforced.

Round 2

Reviewer 1 Report

In regard to the Author´s reply, the following points still remain uncleared:

1) The answer given by authors in regard to negative exergy costs does not seem satisfactory. It is not possible to have negative exergy costs in accordance to the definition of the exergy cost:  B* = B + I and henc ethe unit exergy cost: k*=1+I/B. If you have negative costs, the system violates second law of thermodynamics. The problem may come then from an incorrect cost allocation. You can review works of exergy costs analysis applied to refrigeration system, where in principle you´d get negative exregy costs, but this is due to the incorrect interpretation of cost allocation.

2) In regard to the validation, it was expected that authors would have included the percentage of error or deviation from your model with respect to the design thermal balance (at least of the most important parameters and variables of operation). Or was it zero error?.

Author Response

Reviewer #1: ( "Please see the attachment.")

Thank you for these observations:

Point 1: 1) The answer given by authors in regard to negative exergy costs does not seem satisfactory. It is not possible to have negative exergy costs in accordance to the definition of the exergy cost: B* = B + I and henc ethe unit exergy cost: k*=1+I/B. If you have negative costs, the system violates second law of thermodynamics. The problem may come then from an incorrect cost allocation. You can review works of exergy costs analysis applied to refrigeration system, where in principle you´d get negative exregy costs, but this is due to the incorrect interpretation of cost allocation.

Response 1: In the criterion C1, the residual physical exergy of the exhaust gases discharged, to the environment from the stack, is assumed to the formed within all the components of the gas turbine and HRSG, through which the air or combustion gases passes, that is, on the gas side of the combined cycle.

For this criterion, the residue costs allocation ration for ith productive component involves in the formation of the residue is the ratio of the entropy change of the component to the entropy of the residue.

The entropy changes in the HRSG components on the gas side are negative, however, these components also participate in the formation of the vapor cycle residue in which their entropy changes on the water side are positive. Thus, according to the second law of thermodynamics, the global entropy changes of each of the components of the HRSG are positive.

Since the entropy changes of the HRSG components are negative, then the imputation coefficients of these components to the formation of Eg13* are also negative and therefore the exergy costs.

Although the negative exergy costs lack physical sense, in this case, they only indicate that they reduce the cost of formation of the residue that in this case are the exhaust gases, however, the cost of formation of this residue is always positive. In addition, the following is true for each component of the HRSG:

Ri* = Eg13,i* + EQcond, i* with i = economizers, evaporators, superheaters

where Ri* > 0.

Besides, negentropy was defined as the negative variation of entropy multiplied by the temperature of the environment, when negentropy is used as a fictitious flow to assign costs, inconsistencies may appear in the calculation of the costs, since unit exergy costs may appear with values less than unity. That can happen because the exergy loss is considered as a resource and negentropy as a product, Santos et al. 2009 [*]. In order to overcome those difficulties, we proposed the criterion C2, that is, the residue costs allocation ration for ith productive component involves in the formation of the residue is the ratio of the generated irreversibility of the component to the generated irreversibilities along the process.

[*] Santos, J.; Nascimento, M.; Lora, E.; Martínez, A. (2009). On the Negentropy Application in Thermoeconomics: A Fictitious or an Exergy Component Flow?. Int. J. of Thermodynamics. 12 (4): 163-176.

The following table shows some applications of this criteria to different energy systems where there are the relations of distribution of the cost of residue are negative:

Reference

Study system

Exergy Cost

Explication

- Seyyedi, S. M.; Ajam, H.; Farahat, S. A new criterion for the allocation of residues cost in exergoeconomic analysis of energy systems. Energy 2010, 35: 3474-3482.

- Torres, C; Valero A.; Rangel V.; Zaleta A. On the cost formation process of the residues. Energy 2008, 33: 144-52.

A simple combined cycle

In the criterion of allocate the cost of residues proportional to the entropy generation along the process, for the output gases, the residue cost distribution ratios in superheater, boiler and economizer are negatives

This allocation works for closed cycle, like Rankine or refrigeration cycles, but it fails for other types of process like gas turbine. Since in closed cycles the sum of the entropy generated in each productive process is equal to the entropy saved on the dissipative process, therefore it is logical to distribute the cost of the waste proportional to the entropy generation. But in case of open cycles it is not true. For example, in the case of a simple gas turbine with a heat recovery steam generator, this process saves only a part of the entropy generated in the global process.

[1] Frangopoulos CA (1987) Thermo-economic functional analysis and optimization. Energy 12:563–571.

[2] Lozano MA, Valero A (1993) Thermoeconomic analysis of a gas turbine cogeneration system. In: Richter HJ et al (eds.) ASMEWAM Book No H00874 AES-Vol 30. New Orleans, p. 312–320.

[3] Von Spakovsky MR (1994) Application of engineering functional analysis to the analysis and optimization of the CGAM problem. Energy 19:343–364

A gas turbine cogeneration system

Negentropy cost in heat recovery steam generation

The negentropy flow was applied in thermoeconomics joined up with exergy flow [1]. Negentropy was defined as the negative variation of entropy multiplied by the temperature of the environment. This application represented a great advance in the discipline, since it allowed one to quantify the condenser product, which was not possible before because the condenser is a dissipative component, whose product cannot be expressed in terms of exergy. The concept of negentropy was also used in order to define the productive structure of a gas turbine cogeneration system by Lozano and Valero [2] and Von Spakovsky [3].

Santos, J.; Nascimento, M.; Lora, E.; Martínez, A. (2009). On the Negentropy Application in Thermoeconomics: A Fictitious or an Exergy Component Flow?. Int. J. of Thermodynamics. 12 (4): 163-176.

A steam cycle plant

Negentropy cost in condenser

The introduction of the negentropy in thermoeconomics represented a great advance in the discipline, since this magnitude allows quantifying the condenser product in a steam cycle plant, which was not possible before because the condenser is a dissipative component, whose product cannot be expressed in terms of exergy.

Once the negentropy is the negative of entropy, the equipments that decrease the working fluid entropy are negentropy producers, and those that increase the entropy of the working fluid are negentropy consumers.

2) In regard to the validation, it was expected that authors would have included the percentage of error or deviation from your model with respect to the design thermal balance (at least of the most important parameters and variables of operation). Or was it zero error?.

Response 2: The percentage of error or deviation of some parameters of the combined cycle of our model with respect to the design thermal balance is summarized in the following table:

Parameters and variables of operation

Design thermal balance

Our thermodynamic model

Percentage of error or deviation

Lower Heat Value (kJ/kgf)

41,868

49,494

18.21%

Combined cycle power (kW)

414,000

412,880

0.27%

Gas temperature at the outlet of the gas turbine (°C)

569

617.58

8.53%

Flue Gas Flow Rate (kgcg / h)

1,934,400

2,311,848

19.21%

Reviewer 2 Report

All requirements were met by authors in the revised version of the manuscript.

Author Response

Reviewer #2:

Thank you for your observations.